# Evaluation of animal and plant diversity suggests Greenland's thaw hastens the biodiversity crisis

Carolina Ureta [1,2,7✉], Santiago Ramírez-Barahona [3,7], Óscar Calderón-Bustamante [1], Pedro Cruz-Santiago[1], Carlos Gay-García[1], Didier Swingedouw [4], Dimitri Defrance[5] & Angela P. Cuervo-Robayo [6✉]

Rising temperatures can lead to the occurrence of a large-scale climatic event, such as the melting of Greenland ice sheet, weakening the AMOC and further increasing dissimilarities between current and future climate. The impacts of such an event are still poorly assessed. Here, we evaluate those impacts across megadiverse countries on 21,146 species of tetrapods and vascular plants using the pessimistic climate change scenario (RCP 8.5) and four different scenarios of Greenland's ice sheet melting. We show that RCP 8.5 emission scenario would lead to a widespread reduction in species' geographic ranges (28–48%), which is projected to be magnified (58–99%) with any added contribution from the melting of Greenland. Also, declines in the potential geographical extent of species hotspots (12–89%) and alterations of species composition (19–91%) will be intensified. These results imply that the influence of a strong and rapid Greenland ice sheet melting, resulting in a large AMOC weakening, can lead to a faster collapse of biodiversity across the globe.

[1] Instituto de Ciencias de la Atmósfera y Cambio Climático, Universidad Nacional Autónoma de México, Investigación Científica s/n, C.U., Ciudad de México 04510, México. [2] CONACyT, Consejo Nacional de Ciencia y Tecnología, Av de los Insurgentes Sur 1582, Ciudad de México 03940, México. [3] Departamento de Botánica, Instituto de Biología, Universidad Nacional Autónoma de México, Circuito Exterior s/n, Ciudad de México 04510, México. [4] Environnements et Paléoenvironnements Océaniques et Continentaux, CNRS, Université de Bordeaux, 33615 Pessac, France. [5] The Climate Data Factory, 12 Rue de Belzunge, 75010 Paris, France. [6] Comisión Nacional para el Conocimiento y Uso de la Biodiversidad (Conabio), Insurgentes Sur-Periférico 4903, Parques del Pedregal, Alcaldía Tlalpan, 14010 Ciudad de México, México. [7] These authors contributed equally: Carolina Ureta, Santiago Ramírez-Barahona. ✉email: carolinaus@atmosfera.unam.mx; ancuervo@gmail.com

Rising global temperatures are having negative impacts on biodiversity, increasing the risk of species extinctions across the world[1–4]. If current tendencies of increasing global mean temperatures continue, there is a growing potential of catastrophic, large-scale singular events occurring, such as the melting of Arctic ice sheets[5–7]. A substantial melting of Greenland's ice sheets would generate an additional input of freshwater into the North Atlantic, leading to a substantial rise in sea level and the weakening (or even complete shut-down)[8,9] of the Atlantic Meridional Overturning Circulation (AMOC)[6]—a key element of the global climate system. The potential melting of Greenland's ice sheets is expected to weaken the AMOC, which is responsible for a large amount of meridional heat transportation, resulting in a deceleration of climate warming, but increasing dissimilarities with regional climates[9–12]. These climatic dissimilarities can further grow the probability of species extinctions and ecosystem collapse[7,13].

Recent evidence suggests that the rate of ice-sheet loss over Greenland has accelerated over the last century[5,14–16] and that the AMOC is currently the weakest it has been over the last millennium[15]. Accordingly, the Intergovernmental Panel on Climate Change (IPCC) has highlighted the need to incorporate large-scale singular events into biodiversity risk assessments[7]. Even when the occurrence of such events is not certain[17], there is a non-zero probability of a substantial disruption of the AMOC over the next century, as highlighted by the medium confidence from 2021 IPCC report[18]. Consequently, there is a pressing need for a proper assessment of the likely impacts that this event will have on global biodiversity, which remains scarcely studied[19,20].

To date, only one study has evaluated the impacts of a weaker AMOC on biodiversity, using specifically designed melting experiment scenarios and an ecological niche modeling approach on amphibians across the entire world[20]. This study predicted severe and widespread amphibian declines under a high-emission climate change scenario (RCP 8.5); these declines are largely amplified by a weaker AMOC. Amphibians are often used as bioindicators for environmental change due to their intrinsic sensitivity;[21,22] consequently, it is important to expand the results from this former study and test whether the predicted negative impacts on amphibians could also be observed across all groups of tetrapods and vascular plants (i.e., amphibians, birds, mammals, reptiles, ferns, flowering plants, gymnosperms, and lycophytes). Geographically, we focus on the twelve most biodiverse countries in the world (megadiverse countries) given their global importance of fostering a large number of endemic species, sometimes in relatively small geographic areas, and having high levels of ecological heterogeneity—from tropical forests to desert shrublands[23]. These twelve countries (i.e., Australia, Brazil, China, Colombia, Ecuador, India, Indonesia, Madagascar, Mexico, Peru, Philippines, and Venezuela) collectively harbor about two-thirds (~60%) of the Earth's species of tetrapods and vascular plants[23], and contain most of the world's biodiversity hotspots that are a critical priority for conservation[24]. We decided to work with tetrapods and vascular plants because they have been identified, classified, and studied for a longer period of time than some other groups, such as fungi[25,26]—which have greater taxonomic controversies[26]—and consequently there is more geographic information available[27,28].

We aim to provide a general snapshot of the likely ecological impacts and species exposure under an emission scenario RCP 8.5, considering a control simulation without any Greenland melting, and four sensitivity simulations where Greenland melting estimates are prescribed throughout the simulations. More specifically, we constructed niche-based species distribution models (see "Methods") for 21,146 endemic species to any of the twelve megadiverse countries (Fig. 1). Our projections are focused on climate change impacts and do not consider the likely added effects on species persistence associated with dispersal limitations driven by changes in land-use cover or biotic interactions[29,30]. In addition, our modeling only considers macro-climatic variables, even when other environmental and biological variables are highly relevant for species persistence, especially at small geographic scales. We used the well-known 19 bioclimatic variables available at WorldClim v.2 (Fick 2017)—commonly used in ecological niche modeling—that represent temperature and precipitation annual trends, seasonality, and limiting factors[31]. These variables were used to test if the Greenland ice-sheet melting and the associated changes in AMOC with its corresponding climatic consequences, can affect different biodiversity dimensions, and to what extent in comparison to emission scenario RCP 8.5 (control scenario). The biodiversity dimensions are species distribution (climatic suitability), species richness (SR), differences in species richness (ΔSR), potential hotspots (PHS), and potential composition (estimated using the Sørensen dissimilarity index [$\beta_{SØR}$]).

## Results and discussion

We obtained sixteen ensemble binary maps per species (21,146 animal and plant species): one for the present day and 15 for the five scenarios and the 3 different time horizons (see "Methods"). Our niche-based distribution models projected considerable changes in suitable climatic conditions, which have consequences in geographic ranges, species richness, differences in species richness, and composition. The geographical patterns of biodiversity are greatly altered under the RCP 8.5 scenario, but they are substantially magnified by melting scenarios. Our results suggest that Greenland's thaw, even under the weakest scenario Melting 0.5, is a tipping point for biodiversity, pushing abrupt impacts on species' climatic suitability once this critical threshold is crossed.

In general, species distribution models had a good performance (out of the 21,146 modeled species, 89% had ensemble models with TSS ≥ 0.7, whereas 98% had ensemble models with ROC ≥ 0.85), but with a slightly poorer performance for tetrapods than for vascular plants (Supplementary Fig. 1). This result is probably due to the use of the IUCN polygons to estimate the present-day geographic distribution of tetrapods, which are a broad-scale approximation to species' distributions. Our species distribution modeling was based on a single global circulation model (IPSL-CM5-LR) (see "Methods"), thus inter-GCM variability could not be assessed. However, our models do take into account another important source of modeling uncertainty by using seven different algorithms under an ensemble approach[32] (see "Methods"). The coefficient of variation of our models (reflecting the degree of agreement/disagreement in predictions across algorithms) is in the range of 0.054–0.081, representing robust modeling results (Supplementary Table 1).

**Impacts on geographic ranges.** Our species distribution models project that under RCP 8.5, there would be a widespread contraction in species' geographic ranges (here defined as the geographic extent of species' climatic suitability), but also show that the added climatic alterations under melting scenarios dramatically increase these contractions across countries and groups (Fig. 2, Supplementary Data 1 and 2). More specifically, under RCP 8.5, we estimated the median change in species' geographic ranges, relative to present-day ranges, of −35% (T1: 2030), −59% (T2: 2050), and −78% (T3: 2070) (Fig. 2 and Supplementary Data 2). Under Melting 0.5, which represents a tipping point, the median reduction in species' geographic ranges increases to −95% (T1: 2030), -98% (T2: 2050), and −99% (T3: 2070). Under any melting scenario range, reductions are more pronounced than by control RCP 8.5 scenario (Fig. 2), producing a tipping-

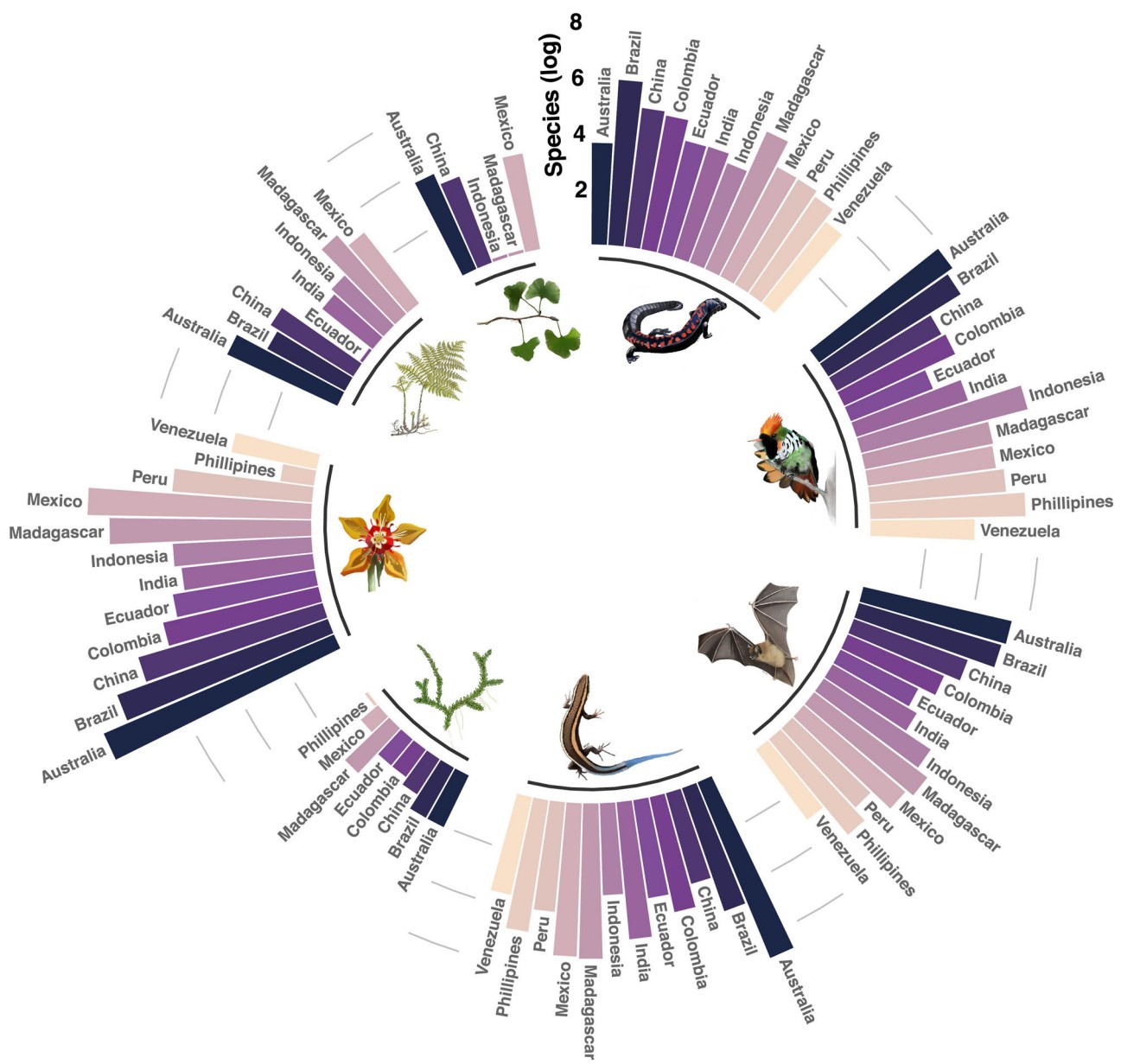

**Fig. 1 Number of species modeled for each group of vascular plants and tetrapods across the twelve megadiverse countries.** Species numbers are given on a log scale. Groups of species are depicted by the inset illustrations (from the top, clockwise): amphibians, birds, mammals, reptiles, lycophytes, flowering plants, ferns, and gymnosperms.

point pattern that has been previously identified for amphibians[20]. Accordingly, our results suggest that amphibians are one of the most at-risk groups under climate change conditions[33,34], but also that this elevated risk extends to other groups of tetrapods and vascular plants across the globe. We also show that on average, vascular plants are expected to be more vulnerable than animals (Fig. 4 and Supplementary Data 3). This might partially be the result of the small number of species modeled for ferns, gymnosperms, and lycophytes. The main reason behind these smaller numbers is the overall diversity of these groups: ~1000 extant gymnosperms, ~1000 extant lyco-phytes, and 10,000 extant ferns. These numbers are substantially smaller than the estimated ~300,000 flowering plants[35,36]. In addition, ferns are believed to have lower levels of endemicity, and gymnosperms are more diverse in northern latitudes and not in the mostly tropical megadiverse countries[37]. There may also be other data biases for these groups that are less studied than their

flowering counterparts. Nonetheless, flowering plants are by far the most abundant group in our dataset—15,162 species—and are nevertheless vulnerable in terms of complete loss of species' geographic ranges (Fig. 3 and Supplementary Data 3). The decline of flowering plant diversity across megadiverse countries will likely increase the extinction risks of other ecologically linked groups[4,38], even when these appear to have an intrinsically lower vulnerability to climate change (i.e., birds; Figs. 2 and 4). The projected substantial negative impacts of climate change on flowering plants raise our concerns about the climate vulner-ability of terrestrial ecosystems due to the likely ecological alterations associated with the decline of these key groups[4,38–40]. Therefore, if current trends of climate change continue, our models imply a probable cascading breakdown of biological interactions, further increasing the probability of species' extinction[41]. The high climatic vulnerability of flowering plants—which are the ecological basis of most terrestrial ecosystems—can

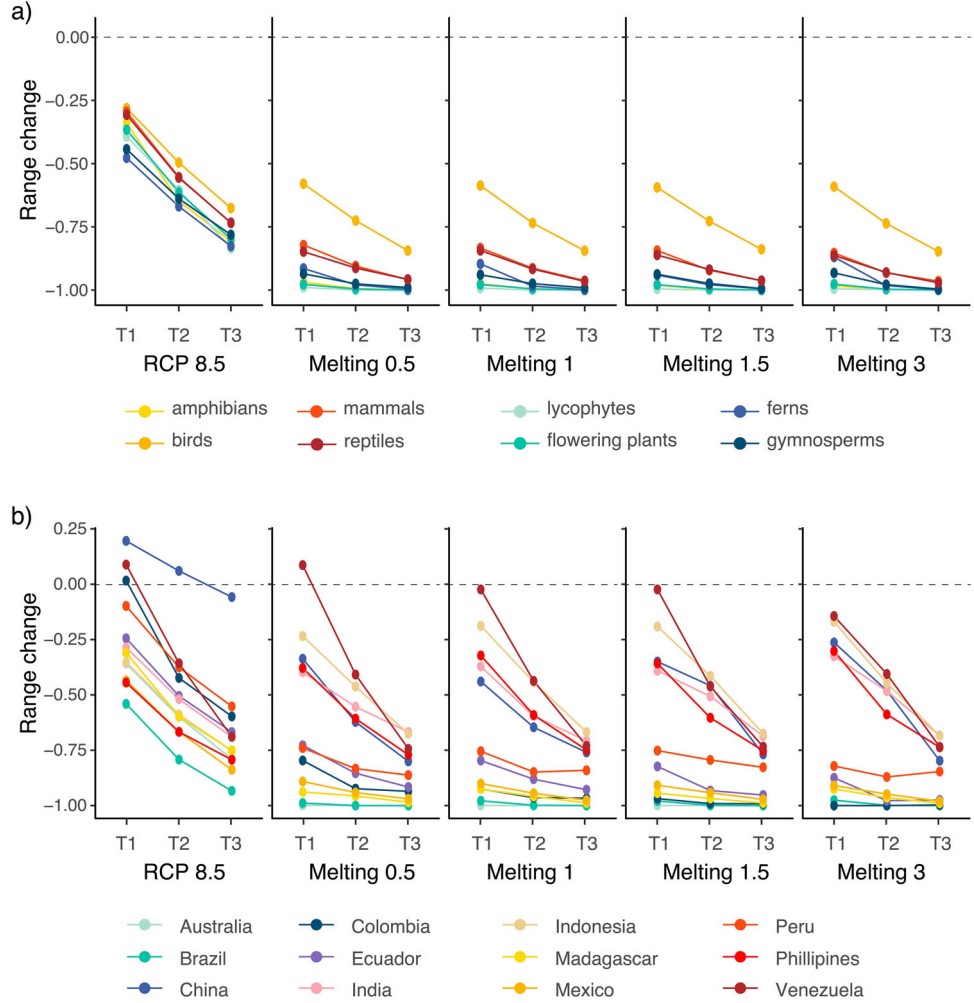

**Fig. 2 Changes in the size of the distribution range of species across taxonomic groups and countries under five scenarios of climate change at three different time horizons. a, b** Estimated median range size change for the eight **a** taxonomic groups and **b** countries across scenarios and time horizons. Species range sizes were standardized relative to the present-day species range size and then summarized across groups: negative values indicate range reductions (1 means complete loss), and positive values indicate range expansions. T1: 2030; T2: 2050; T3: 2070.

potentially increase the collapse of diversity in other groups of plants (epiphytic ferns and bryophytes) and animals (tetrapods and insects), the later including many pollinators, frugivores, and herbivores[39,42,43].

Across countries, the impacts of the melting of Greenland's ice sheet are geographically heterogeneous (Fig. 2, Supplementary Figs. 2 and 3). The median change in species' geographic ranges varies, and in some cases our models project range expansions (Fig. 2 and Supplementary Data 2). Previous studies have reported that for particular geographic regions (e.g., China) and individual species, changes in climatic conditions, such as an increase in mean annual temperature or annual precipitation, might increase their suitability[44,45]. In this context, we show that in some megadiverse countries, species geographic ranges increase under RCP 8.5 by T1: 2030, China (median of 19.6% increase), Venezuela (median of 8.9% increase), and Colombia (median of 1.7% increase) (Fig. 2 and Supplementary Data 1); these increases are reverted to median range losses by T3: 2070 for all three countries. China was the country showing greater increases in geographic ranges under control RCP 8.5, and there are several reasons that might explain this result. China has a higher seasonality than all other countries evaluated (Supplementary Figs. 12–22), and consequently its endemic species might have higher adaptability[46]. In addition, annual temperature in

China is also colder than most of the other countries evaluated and an increase in temperature might be beneficial for some species[44]. China is a country with diverse landforms, including mountains, plateaus and hills that account for ~70% of its topography, having large areas with very high altitude (>3000 m asl)[44]. Species from high altitudes are regularly vulnerable to climate change because they cannot seek cooler temperatures at higher elevations if conditions get warmer[40,45,46]. In China, under a full dispersal scenario, species can move to higher altitudes if temperatures increase (Fig. 6 and Supplementary Fig. 2). However, changes in temperature and precipitation can only be partially associated with declines in species' geographic ranges; in Madagascar, for example, changes in temperature and precipitation (annual trends and seasonality) appear to be similar under RCP 8.5 and Melting 0.5 (Supplementary Figs. 10, 11 and 18–19).

Our results also show that even under the "weakest" melting scenario, Melting 0.5, strong reductions in the geographic areas of species' climatic suitability are expected. Even when melting scenarios encompass a slowdown in global warming (Supplementary Figs. 4–11), they also entail greater changes to the present-day scenario than RCP 8.5—i.e., larger differences in maximum temperature of warmest month and precipitation of driest month—(Supplementary Figs. 12–19). The generally more drastic climate alterations projected under melting scenarios

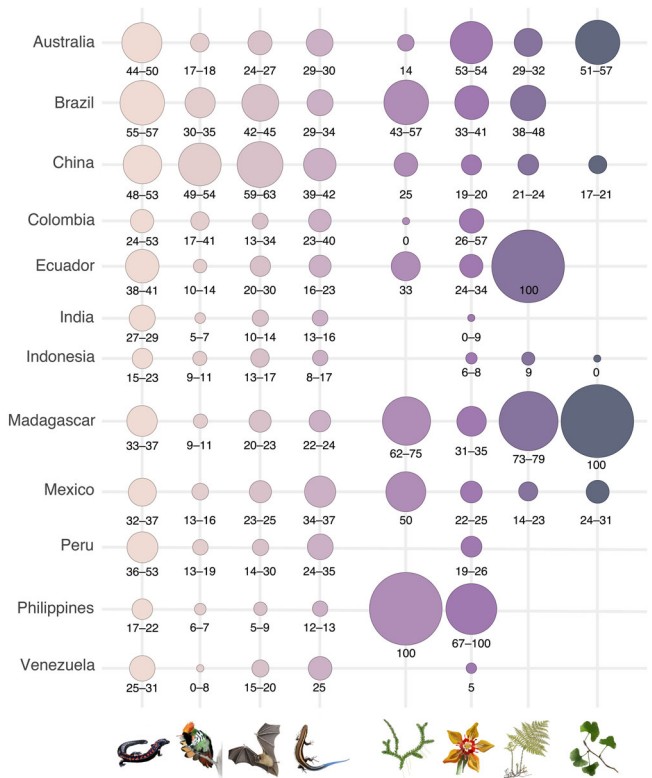

**Fig. 3 Proportion of species of vascular plants and tetrapods with complete range loss across twelve megadiverse countries.** The proportion of species (%) with complete range loss was estimated relative to the total number of species modeled for each group and country. The numbers below circles indicate the range of values estimated for 2030 across the four melting scenarios. Range size estimation was based on species distribution models and was standardized relative to the present-day species range size. Groups of species are indicated by the inset illustration (from left to right): amphibians, birds, mammals, reptiles, lycophytes, flowering plants, ferns, and gymnosperms. Please find time horizons T2: 2050 and T3: 2070, in the Supplementary Material.

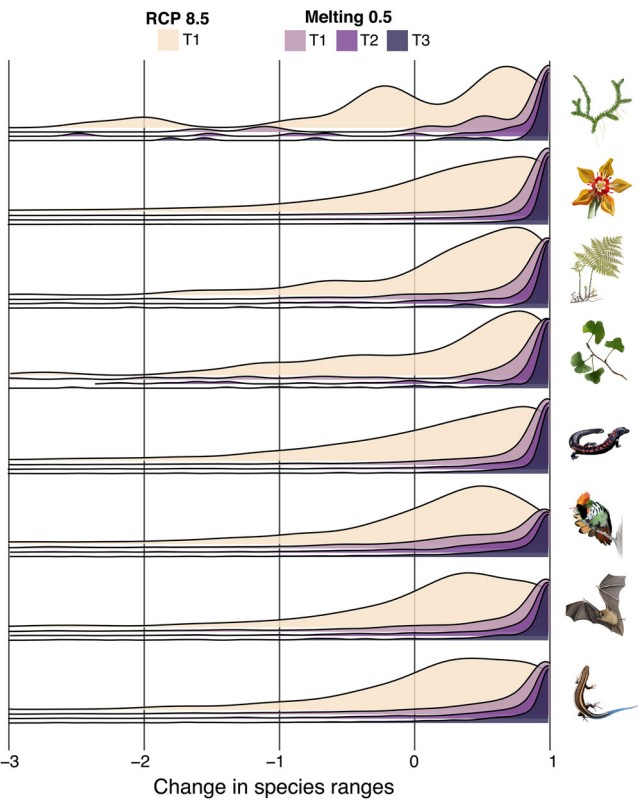

**Fig. 4 Changes in the size of the distribution range of species of vascular plants and tetrapods.** Estimates of range size were based on species distribution models (SDMs). Range sizes were standardized relative to the present-day species range size: positive values indicate range reductions (1 means complete loss), and negative values indicate range expansions. Estimates of range size were aggregated by taxonomic group and visualized using density plots for T1: 2030 under the RCP 8.5 scenario and for T1: 2030, T2: 2050, and T3: 2070 under the Melting 0.5 scenario. Groups of species are indicated by the inset illustration (from top to bottom): lycophytes, flowering plants, ferns, gymnosperms, amphibians, birds, mammals, and reptiles.

(Supplementary Figs. 12–22) might explain the stronger and more homogeneous decrease in climatic suitability observed across megadiverse countries.

Brazil and Australia show the highest numbers of species range loss (complete losses under RCP 8.5 in the range of 7–24% and 8–32% across time horizons, Supplementary Data 4). On the contrary, Colombia and China show a lower proportion of species with complete range loss under RCP 8.5 (for both countries, losses are in the range of 3–8% across time horizons) (Supplementary Data 4). With the added contribution of melting (Melting 0.5), the proportion of species with complete range loss substantially increases across the board (Fig. 2); in Brazil and Australia, these proportions increased to 50–60% and 36–59%, respectively, whereas in Colombia and China these increase to 23–32% and 27–34%, respectively.

Our results are consistent with substantial alterations on climates zones predicted under melting scenarios[10] and imply a considerable impact for the worlds' biodiversity; in the case of South American megadiverse countries, the large-scale singular event of Greenland's thaw will further increase the global concerns on the region's deforestation trends[7,47]. In this context, other anthropogenically-driven factors, such as land-use changes and invasive species[48], could have a synergic effect with climate change that will further impact species persistence and hinder conservation efforts[1,49,50]. On the other hand, island countries,

such as Indonesia and the Philippines, are intrinsically more at-risk to climate-related impacts, including those associated with the sea-level rise due to the melting of polar ice sheets. Indeed, climate risks in these island countries are particularly concerning due to vulnerability to sea-level rise[51] and the fact that the dispersal abilities of terrestrial species (plants and animals) are hindered by the open ocean[52].

Even assuming that species would be able to disperse freely, the modeling under RCP 8.5 shows complete disappearance of climatically suitable areas for 1239 species by T1: 2030 and 4,483 species by T3: 2070 (Fig. 4 and Supplementary Data 3); these represent 5.8% and 21.2% of the total number of modeled species. Hence, there is more than a doubling of the threat of disappearance. In turn, the added contribution of Greenland's thaw under Melting 0.5 pushes these numbers to 7,728 species by T1: 2030 and 10,305 by T3: 2070 (Fig. 4 and Supplementary Data 3), representing 36.5% and 48.7% of the modeled species. In comparison to RCP 8.5, the three melting scenarios are almost equally severe, once again suggesting a tipping-point behavior over biodiversity loss of the countries evaluated (Fig. 5).

**Impacts on species richness (SR), differences in species richness (ΔSR), and composition.** Given that several species have an important decrease in their climatic suitability (geographic

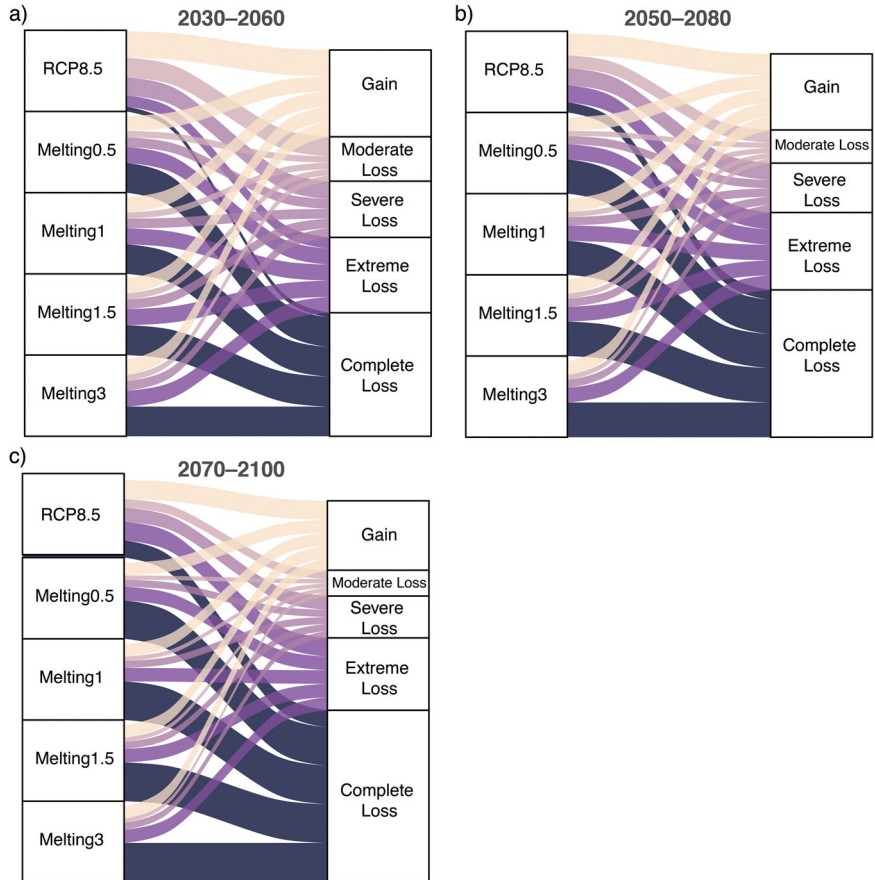

**Fig. 5 Proportion of species falling within different categories of change in species' range sizes under different climate change scenarios. a–c** Alluvial plots showing the distribution of five categories of change in species' range size under the RCP 8.5 and the four melting scenarios. The vertical size of the blocks and the width of the flows are proportional to the frequency of species within each block/flow. All scenarios have the same block size corresponding to the 21,146 modeled species. The flows represent the proportion of species within any of the five categories (each of which has a color) as estimated under each of the five scenarios. Range size estimation was based on species distribution models and was standardized relative to the present-day species range size.

ranges) under all scenarios evaluated, areas that were climatically suitable for several species will be suitable for fewer species (Fig. 6). Species richness was defined as areas where a larger number of species have climatic suitability (SR); whereas difference in species richness (ΔSR) results from subtracting two different species-rich maps (see "Methods"). In all countries evaluated, except for China, the highest rich geographic areas will decrease from the present-day scenario to RCP 8.5 and will remarkably decrease with melting scenarios (Figs. 2 and 5). Once more, China remains an exception, and as explained above, we think this result might be related to a greater seasonality of the country that might increase its endemic species adaptability[53], its more temperate climate in comparison to the other countries evaluated[54], and the fact that it has a very large area with high altitudes[44]. Decreases in species-rich areas might have repercussions in conservation strategies[24].

Maps of the differences in species richness (ΔSR) gives insight of the shifts in climatic suitability for the species evaluated. These maps give complementary information to species-rich areas, identifying aspects of biodiversity change that are decoupled from species richness, such as potential species turnover with their corresponding ecological consequences[55]. Potential species turnover in a geographic area can be given by an increase or a decrease in species climatic suitability. All countries evaluated show areas with gains and losses in RCP 8.5, but in the melting scenario, areas with gains are quite scarce except for China

(Fig. 6). However; species turnover can be expected either by gains or losses.

We also evaluated potential species hotspots (PSH) that were defined as those regions with the highest level of species richness observed in the present-day within each country. In this case, we used a threshold that identified grid cells with a species richness greater than the maximum SR*0.6 (see "Methods"). Not surprisingly and given our selection of endemic species, the PSH coincide with globally important biodiversity hotspots[24], which harbor an important percentage of the endemic and threatened species of the world. More than twenty years ago, it was estimated that the efficacy in protecting these biodiversity hotspots that collectively encompass less than 2% of the Earth's surface would translate into the protection of 44% of vascular plants and 35% tetrapods. Our results show that under RCP 8.5, by T1: 2030, reductions in PSH are in the range of 12–91% across countries; for instance, under this scenario, the Brazilian PSH covers an area equivalent to 9% of the present-day PSH, whereas the Peruvian and Colombian PSH encompass an area equivalent to 88%. Overall, these reductions are magnified with the added contribution of Greenland's melting, where the shrinking of PSH rises to 40–100% by T1: 2030 under Melting 0.5 (Fig. 7, Supplementary Data 5, and Supplementary Figs. 28–33); this highlights the tipping-point impacts of Greenland's ice sheets melting on the world's biodiversity hotspots. However, some countries show an increasing extent of PSH under climate change

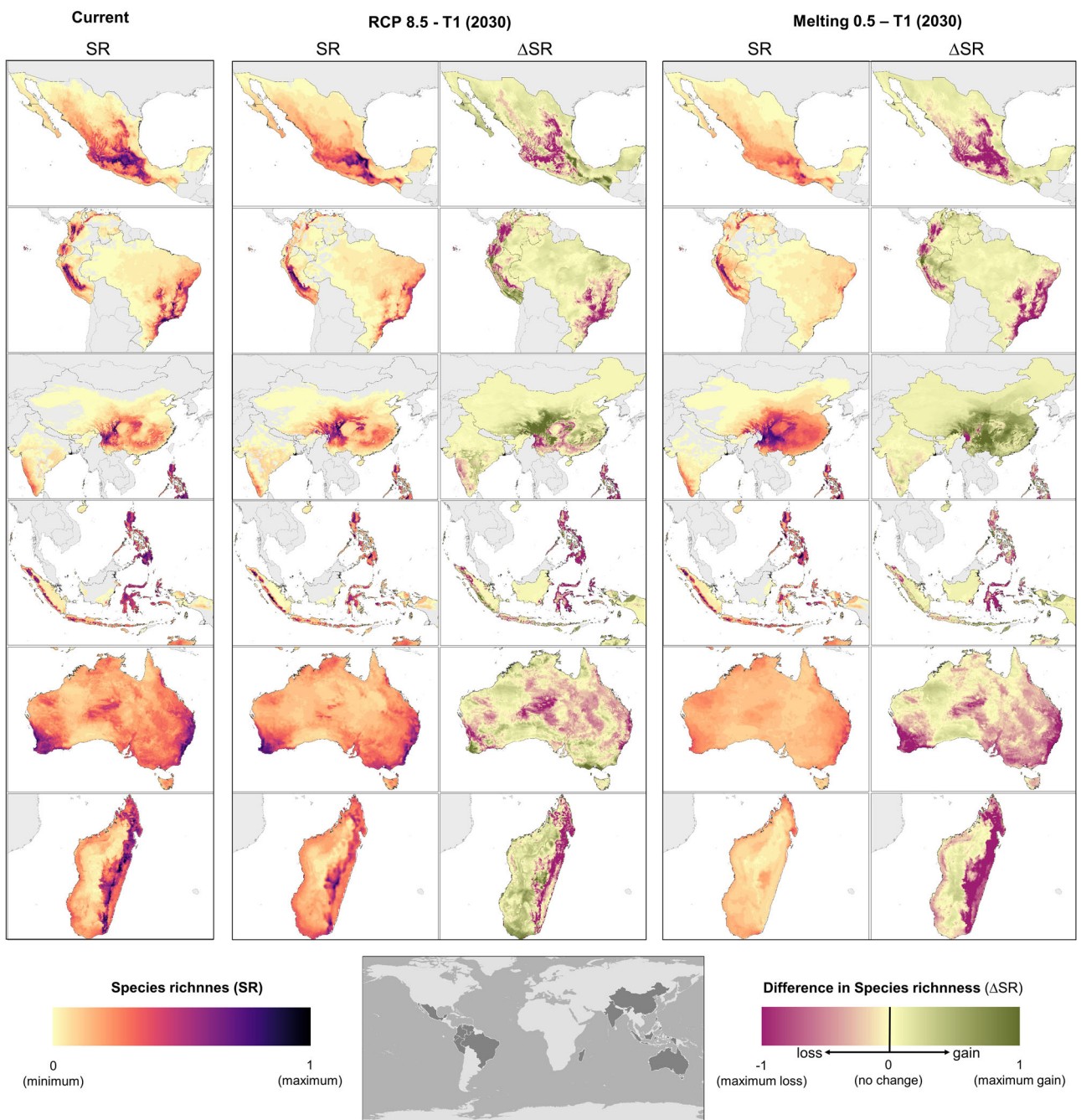

**Fig. 6 Geographic patterns of present-day species richness and temporal changes in species richness across twelve megadiverse countries.** Estimates are based on species distribution models (SDMs) of vascular plants and tetrapods. Species richness (SR) was standardized to the range 0–1 within each country. Differences in species richness (ΔSR) are shown for T1: 2030 under Melting 0.5 scenario. For each grid cell, the change in species ranges was estimated as the difference in the number of species that find suitable climatic conditions in a specific geographic area and scenario. Please find the separated map for animals and vascular plants in the Supplementary Material.

scenarios; for RCP 8.5, the Chinese and Venezuelan PSH increases to 153 and 139% by T1: 2030, respectively; for the Chinese PSH, this increasing trend continues through time and holds under Melting 0.5 by T1: 2030 (Fig. 7 and Supplementary Data 5). Importantly, the observed trends in China suggest that the thaw of Greenland's ice sheets might have positive impacts on biodiversity (that is, increasing PSH extent), but these appear to be transient; thus, by T3: 2070, the extent of PHSs across all megadiverse countries is substantially diminished under melting scenarios. More so, even within expanding PSH, there is a

noticeable decrease in species richness, evident in the diminution of the maximum species richness within PSH (Supplementary Fig. 28). In other words, all PSH identified in megadiverse countries are highly vulnerable to climate change, especially to tipping points that will likely push the climate-biodiversity system into a new state[13,56,57]. Thus, although our modeled species represent a small fraction of global diversity, the alterations to the geographic extent of PSH and species' geographic ranges are an alarming possibility. Furthermore, our results also project that PSH would be subjected to moderate to high changes in species

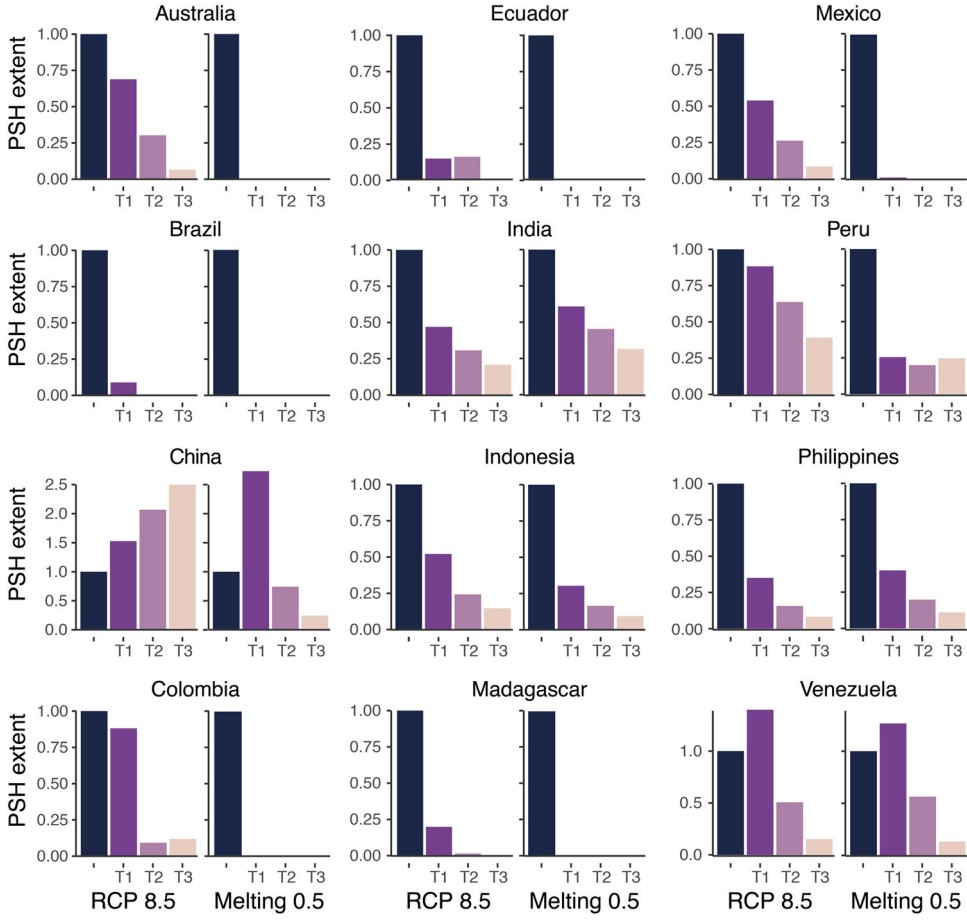

**Fig. 7 Temporal changes in the spatial extent of potential species hotspots (PSH) across twelve megadiverse countries.** The extent of PSH was measured for each country separately as the number of pixels with a species richness (SR) higher than 0.6 × maximum SR. For each country, the extent of PSH was standardized relative to the present-day extent (first column), where values greater than one indicate an expansion of PSH and values of zero indicate the complete disappearance of PSH. T1: 2030; T2: 2050; T3: 2070. To find other melting scenarios please see the Supplementary Material.

composition (Fig. 8). Again, we estimated an increased impact by Greenland's thawing ice sheets—and the ensuing weaker AMOC —on species composition (Fig. 8); for instance, the median temporal dissimilarity for all PSH under RCP 8.5 (T1: 2030 $\beta_{SØR} = 0.361$; T2: 2050 $\beta_{SØR} = 0.511$; T3: 2070 $\beta_{SØR} = 0.641$) is considerably lower than under Melting 0.5 (T1: 2030 $\beta_{SØR} = 0.608$; T2: 2050 $\beta_{SØR} = 0.682$; T3: 2070 $\beta_{SØR} = 0.739$) (Supplementary Data 6).

Based on our models, we suggest a dramatic decline and alteration of biodiversity across megadiverse countries within a relatively short period after the onset of Greenland's melting ice sheets and the ensuing weakening of the AMOC[11]. Thus, our assessment represents an important contribution to the evaluation of biodiversity on a global scale under the possibility of this large-scale singular event. The projected reduction and complete loss of climatically suitable areas for a great number of plants and animals, with the ensuing decline in species richness and changing composition across all megadiverse countries, highlight the threat to biodiversity posed by ongoing climate change. In this case, the risks to endemic species are of paramount concern because if these fail to respond to climate change, by either adapting or migrating, they will likely go extinct. In addition, extreme weather events, such as hurricanes, droughts, and fires, which are expected to increase in frequency and intensity under current climate change[18], will further push the risk of species going locally and globally extinct[58,59]. Our niche-based distribution models for animals and plants suggest that the projected

degradation of endemic biodiversity within megadiverse countries is pushed to collapse by the additional contribution by the thawing of Greenland's ice sheets, which is usually neglected in climate change simulations[20]. In light of recent observations of substantial ice-sheet loss in Arctic regions and the fact that the AMOC is currently at its weakest point in millennia[14,15], our projections provide reasons of major concern for the future of endemic species across the world's most biodiverse countries.

## Methods

**Species occurrence records.** We compiled data on the distribution of 21,252 endemic species of any of the twelve megadiverse countries from four tetrapod (5,757) and four vascular plant groups (15,389) (amphibians, reptiles, birds, mammals, lycophytes, ferns, gymnosperms, and flowering plants). Species occurrence records were obtained from the Global Biodiversity Information Facility (GBIF)[27], the International Union of Conservation of Nature (IUCN)[28], and BirdLife[60,61]. We only modeled species with at least 25 unique records at a 5 arc-minute resolution (~10 km at the equator). In many cases, the processing of the IUCN polygons resulted in species with thousands of occurrence records. In these cases, we randomly chose a maximum of 500 records per species. The greater the number of observed records, more problems can be associated with spatial bias in the modeling[62]. In the case of records coming from IUCN polygons, more records require more computing time and these do not necessarily provide more information into the modeling given that their distribution is quite homogeneous.

For tetrapods, we first explored the possibility of using occurrence records from GBIF, but data for megadiverse countries were scarce. Consequently, we decided to use the distribution polygons provided by the IUCN for amphibians, reptiles, and mammals (terrestrial and freshwater species)[28], and the distribution polygons provided by BirdLife[60]. We based this decision on the fact that ecological niche modeling using IUCN polygons has been proven to give robust results[20]. For the

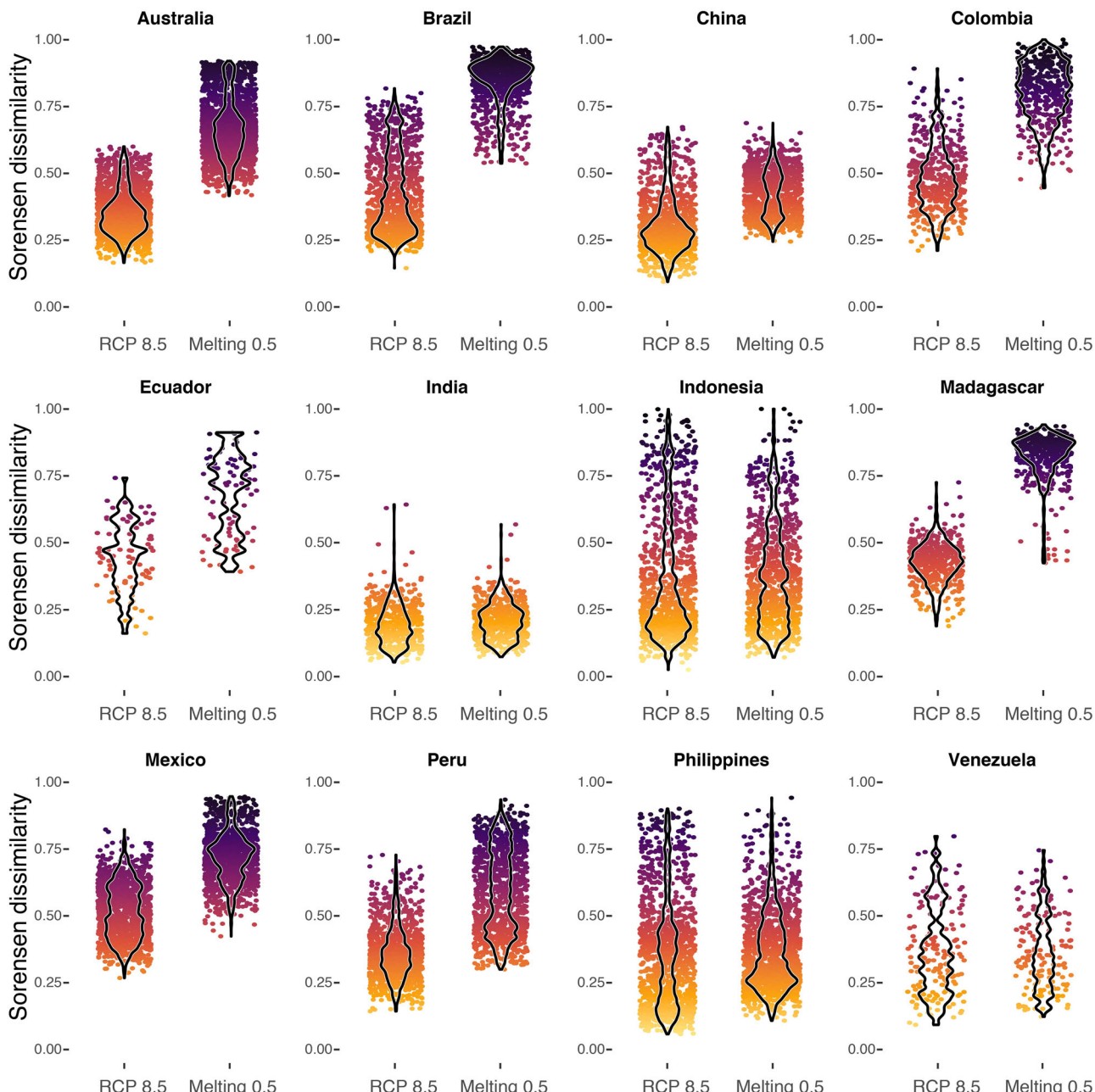

**Fig. 8 Changes in species composition within potential species hotspots (PSH) across the twelve megadiverse countries.** Temporal changes in species composition were based on species distribution models (SDMs) and were estimated using the Sørensen dissimilarity index ($\beta_{S\varnothing R}$) for individual pixels across time. Values approaching one indicate increasing dissimilarity in composition across time. Changes in composition are shown across countries for T1: 2030 under the RCP 8.5 and Melting 0.5 scenario.

IUCN polygons, we retained species that have been categorized as "extant", "possibly extinct", "probably extant", "possibly extant", and "presence uncertain", discarding species considered to be "extinct". In addition, we did not model species reported by the IUCN as "introduced", "vagrant", or those in the "assisted colonization" category; for mammals and birds, we only considered the distribution of "resident" species. Depending on the taxonomic group, and given the information available, we used different approaches to identify species endemic to any of twelve megadiverse countries: Australia, Brazil, China, Colombia, Ecuador, India, Indonesia, Madagascar, Mexico, Peru, Philippines, and Venezuela. For birds, we used BirdLife to identify species listed as "breeding endemic" and then choose the corresponding IUCN polygons. To identify the rest of endemic species in the other groups, we used a 0.08333° buffer around each country to select the IUCN polygons that fall completely within the country limits. We converted all selected species polygons into unique records at a 5 min resolution (~10 km at the equator).

For vascular plants, we used geographic occurrence data obtained from the Global Biodiversity Information Facility by querying all records under "Tracheophyta" (we only considered "Preserved Specimens" in our search). Plants records were

taxonomically homogenized and cleaned following the procedures described in ref. [63] using Kew's Plants of the World database[64] as the source of taxonomic information. Mostly, we identified endemic species as those with all occurrence records restricted to any given megadiverse country. For countries in which data for vascular plants were scarce or absent (e.g., India), we complemented occurrence information with polygons from the IUCN (although IUCN data for plants remains limited) following the procedure described for tetrapods.

**Climatic data**. We used the 19 bioclimatic variables available at WorldClim v.2 (Fick 2017) as the baseline (present-day) climatic conditions (1970–2000) (annual mean temperature, mean diurnal range, isothermality, temperature seasonality, the maximum temperature of the warmest month, minimum temperature of the coldest month, temperature annual range, mean annual range, mean temperature of wettest quarter, mean temperature of driest quarter, mean temperature of warmest quarter, mean temperature of coldest quarter, annual precipitation, precipitation of wettest month, precipitation of driest month, precipitation seasonality,

precipitation of wettest quarter, precipitation of driest quarter, precipitation of warmest quarter and precipitation of coldest quarter). From this baseline scenario, bioclimatic variables start to vary because of climate change. We used bioclimatic variables derived from the IPSL-CM5-LR ocean-atmospheric model under five scenarios: (i) the high-emissions RCP 8.5 W/m²; and (ii) melting scenarios consisting of four different experiments of freshwater discharge into the North Atlantic from Greenland's meltwater (see DeFrance[16] for details). We acknowledge that using a single GCM does not allow us to estimate inter-GCM variability in the resulting distribution models; however, the melting scenarios do only exist for IPSL-CM5-LR GCM. We applied as control scenario RCP 8.5 because melting scenarios would have been more complicated to support with lower emission scenarios. In addition, we are using well-designed opportunity experiments from ref. [11] and wanted to be consistent with their choice of RCP 8.5. Also, these experiments are based on CMIP5, which shows similar climate impact fingerprints than CMIP6[65]. This might be explained by the fact that CMIP5 and CMIP6 are still relatively close, and that the main climatic effects of the AMOC are already well-represented by the climate dynamics in CMIP5.

The four melting scenarios are equivalent to a sea-level rise of 0.5, 1.0, 1.5, and 3.0 meters above the current sea level, and these are named accordingly: Melting 0.5, Melting 1.0, Melting 1.5, and Melting 3.0. These AMOC scenarios are experiments that were superimposed to the RCP 8.5 scenario adding 0.11, 0.22, 0.34, and 0.68 Sv (1 Sv = 106 m³/s) coming from a freshwater release that starts in 2020 and finishes in 2070 (Anthoff et al.[14]). We obtained debiased bioclimatic variables[11] under the five future scenarios for three consecutive time horizons: T1: 2030 (2030–2060); T2: 2050 (2050–2080); and T3: 2070 (2070–2100). The time horizons evaluated represent short, medium, and long terms in order to help decision-makers order conservation priorities.

**Ecological niche modeling**. At their most basic, the algorithms used to construct species distribution models relate species occurrence records with climatic variables to create a climatic profile that can be projected onto other time periods and geographic regions[66]. The resulting models have proven useful in evaluating the impacts of climate change on biodiversity and to identify varying levels of vulnerability among species[32,67,68]. Here, we employed a multi-algorithm (ensemble) approach to construct species distribution models as implemented in the "biomod2" package[67] in R[69] (Supplementary Fig. 33). The underlying philosophy of ensemble modeling is that each model carries a true "signal" about the climate-occurrence relationships we aim to capture, but it also carries "noise" created by biases and uncertainties in the data and model structure[32,67]. By combining models created with different algorithms, ensemble models aim at capturing the true "signal" while controlling for algorithm-derived model differences; therefore, model uncertainty is accounted for during model construction (see Supplementary Material for further detail).

Prior to modeling, we reduced the number of bioclimatic variables per species by estimating collinearity among present-day bioclimatic variables. We employed the "corrSelect" function of the package fuzzySim[70] in R[69], using a Pearson correlation threshold of 0.8 and variance inflation factors as criteria to select variables. Given the number of species evaluated and the ecological information scarcity, we did not select a set of variables based on ecological knowledge by each of the species modeled. Instead, for the variables pre-selection, we used the statistical approach described above that has been proven to give models with good performance[71,72]. We used seven algorithms with a good predictive performance (evaluated with the TSS and ROC statistics; Supplementary Fig. 1): Maxent (MAXENT.Phillips), Generalized Additive Models (GAM), Classification Trees Analysis (CTA), Artificial Neural Networks (ANN), Surface Range Envelope (SRE), Flexible Discriminant Analysis (FDA), and Random Forest (RF). Because occurrence datasets consisted of presence-only data, for each model, we randomly generated 10,000 pseudo-absences within the model calibration area; we gave presences and absences the same importance during the calibration process (BIOMOD's prevalence = 0.5). For each species, we selected a calibration area (i.e., the accessible area or M)[73] using a spatial intersection between a 4° buffer around species occurrences and the terrestrial ecoregions occupied by the species[73] (Supplementary Fig. 33). The projected M (i.e., the area accessible for species in future scenarios) was defined using a 2° buffer around the present-day calibration area (M). By limiting the M, we incorporated information about dispersal and ecological limitations of each species into the modeling[66]. We did this to take into account a more realistic dispersal scenario given the velocity with which climatic changes are happening and because there are geographic and ecological barriers, which is the reason why we used ecoregions to limit our M. We assumed climatic niche conservatism across time; and inside the projected M we also assumed full dispersal. Consequently, inside the projected M, the evaluated species can win or lose suitable climatic conditions.

We calibrated each algorithm using a random sample of 70% of occurrence records and evaluated the resulting models using the remaining 30% of records. To validate the predictive power of the ecological niche models, we used the True Skill Statistics (TSS) and the Receiver Operating Characteristics (ROC) and performed 10 replicates for every model, providing a tenfold internal cross-validation. To account for uncertainty, we constructed the ensemble models (seven algorithms × ten replicates) using a total consensus rule, where models from different algorithms were assembled using a weighted mean of replicates with an

evaluation threshold of AUC > 0.7 (Supplementary Fig. 1). However, as shown by the distribution of validation statistic in Supplementary Fig. 1, most ensemble models presented a very good predictive power (AUC > 0.8). In some cases, modeling issues in some insular species required that we change the calibration area (M) to the entire country.

We used the resulting ensemble models to project the potential distribution of each species under both current and future climatic conditions (Supplementary Fig. 34). We then examined the frequency in which different bioclimatic variables appeared to have the highest contribution during model construction for each species. The algorithms used (Maxent, GAM, CTA, ANN, SRE, FDA, and RF) identify these variables by iteratively testing combinations of all the available variables (i.e., those selected based on low correlation values) until reaching a set of variables that was most informative on the distribution of species; this set of variables had the highest predictive power of species occurrence. For every species, we retrieved the two variables with the largest model contribution (Supplementary Figs. 34 and 35).

**Species geographic range**. We converted ensemble probability maps into binary maps of presence/absence using the TSS threshold; these binary maps reflect the distribution of climatic suitability of species, where values of 0 and 1 represent grid cells with non-suitable and suitable climates, respectively. In order to approximate the vulnerability of individual species to climate change, we estimated the temporal changes in the extent of the area of climatic suitability (geographic range) for every species relative to the present-day distribution. We estimated species' geographic ranges by identifying and counting those grid cells with suitable climatic conditions (values of 1) in the present-day and under future scenarios. We then estimated the proportion of range changes through time, quantifying the proportion of grid cells either lost or gained for each species. This allowed us to estimate the proportion of species (by country and group) projected to have a complete loss of geographic ranges in the future.

**Species richness, differences in species richness, potential species hotspots (PSH), and temporal dissimilarity**. We used binary maps to construct presence-absence matrices (PAM), which contain information on the presence (values of 1) or absence (values of 0) of species across grid cells. Using these PAMs, we estimated species richness (SR) as the sum of species present in each grid cell; to visualize SR across space, we generated 16 species richness maps corresponding to the present-day and the four future scenarios at each of the three temporal horizons. We used these maps to estimate and visualize temporal differences in species richness (ΔSR) over time by subtracting the estimated SR in the future from the current SR, for every grid cell; for visualization, we standardized SR per country to the range 0–1. We assumed full dispersal ability of species in all analyses, meaning that all suitable areas in the future had the same probability of being occupied, irrespective of the distance to the present-day distribution.

By calculating species richness (SR) across grid cells, we defined Potential Species Hotspots (PSH) within each country as those grid cells with the highest levels of SR. For this, we defined the PSH by calculating the maximum present-day species richness (maxSR) observed in each country and then identified grid cells with richness values above a threshold of maxSR*0.6. Considering only those grid cells with a SR above this threshold, we estimated the geographic extent of PSH across time periods and scenarios and estimated changes to the extent of PSH relative to present-day conditions. Given that we use the threshold to define PSHs, we tested two additional thresholds (20 and 90%) to define and quantify the extent of PSHs. However, these additional results agree with the general trend. We chose not to base our threshold on the distribution of SR values (i.e., quantiles, median) due to the high proportion of grid cells with SR < 10.

For each PSH, we estimated the change in species composition over time using the Sørensen pairwise dissimilarity index (β$_{SØR}$), which estimates the dissimilarity in species composition between two sites and incorporates both turnover and differences in species richness among sites. For this, we estimated dissimilarity between the present-day and each of the three temporal horizons at each spatial location within PSH and summarized dissimilarity values for all PSH scenarios. The observed temporal dissimilarity reflects two main patterns of varying composition under climate change scenarios: (i) the replacement of present-day species by "new" species within sites and (ii) the loss (or gain) of species resulting in nested species assemblages. Values of temporal β$_{SØR}$ approaching one are indicative of higher dissimilarity between the present-day species composition and the future projected composition within sites, and values approaching zero are indicative of few temporal changes in composition.

**Characterization of climate changes**. We characterized the bioclimatic profile across countries to explore the possible influence of different variables (e.g., temperature, precipitation) on the observed changes to species' geographic ranges and richness. For this, we estimated the temporal change in four bioclimatic variables representing annual trends (i.e., mean annual temperature, annual precipitation) and seasonality (i.e., the maximum temperature of the warmest month, precipitation of the driest month) under the RCP 8.5 and Melting 0.5 and across all temporal horizons.

Finally, we explored whether the current climate differs between areas showing declines in species richness and those showing increasing SR. For this, we used the resulting per-country maps of temporal differences in species richness (ΔSR) to identify grid cells with estimated positive and negative ΔSR and then characterized these areas in terms of their bioclimatic profiles. We estimated climatic profiles only for those grid cells with the largest gains and losses in species richness (positive and negative ΔSR), which were defined as grid cells with ΔSR values above the third quantile and below the first quantile of the distribution of ΔSR, respectively. We characterized these areas using four bioclimatic variables: mean annual temperature, temperature seasonality, total annual precipitation, and precipitation seasonality.

**Reporting summary**. Further information on research design is available in the Nature Research Reporting Summary linked to this article.

## Data availability

Data for species distribution models are available at Zenodo with the identifier https://doi.org/10.5281/zenodo.4917258. The geographic occurrence data for vascular plants is available from the Global Biodiversity Information Facility with the identifier https://doi.org/10.15468/dl.bdxzkw. The distribution polygons for tetrapods and vascular plants are available at https://www.iucnredlist.org/ and http://www.birdlife.org/.

## Code availability

All R codes used for processing the distribution models, and to perform the geospatial and statistical analyses are available at Zenodo with the identifier https://doi.org/10.5281/zenodo.4917258.

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

## Acknowledgements

We are grateful to Francisco Estrada-Porrúa, Julián A. Velasco, and the members of "Clima y Sociedad" for the ideas that prompted the development of the project and comments on earlier drafts. Part of the analyses in this paper were carried out on CONABIO's (Comisión Nacional para el Conocimiento y Uso de la Biodiversidad) computing cluster, supported by their system administrator and the Subcoordinación de Soporte Informático; and the other part of the analyses in this paper were carried out on the computing cluster Tláloc (Centro de Ciencias de la Atmósfera) supported by their system administrator.

## Author contributions

C.U. and A.P.C.-R. conceived of the idea with the help of C.G.-G.; C.U., S.R.-B., and A.P.C.-R. designed the research and analyses; C.U. and A.P.C.-R. compiled and processed the data on animals; S.R.-B. compiled and processed the data on plants; D.S. and D.D. provided the climate simulations for melting scenarios; O.C.-B. generated and processed the bioclimate layers used in the modeling; P.C.-S. and A.P.C.-R. processed the data and performed the setup of the computing cluster for the analyses; C.U., S.R.-B., and A.P.C.-R. performed the analyses; C.U. and S.R.-B. led the writing with contributions from A.P.C.-R.; C.U., S.R.-B., D.S., D.D. and A.P.C.-R. discussed the manuscript. All authors read and approved the manuscript.

## Competing interests

The authors declare no competing interests.
