## [Peer Review File · Communications Biology]

Reviewers' comments:

Reviewer #1 (Remarks to the Author):

#Reviewer comments

In this contribution, the authors measured the impact of climate change using a high-emission global warming scenario (RCP 8.5) and four melting scenarios, to model and predict the distribution of 221,146 species in 12 different countries.

The authors used their conclusion to highlight the importance of thermohaline circulation collapse in the future distribution of different taxonomic groups. They showed that melt-ing scenarios magnifies the decline in species geographical extend.

In summary, this contribution is an interesting work; and the results are nicely presented. Nonetheless, I have some major concerns and few very minor observations.

1- The result & Discussion section is poorly written. The result starts with reviewing the method and then jump to the discussion. For example, in line 102 which is the second paragraph of the result & discussion, when the authors mentioned "our results agree with previous analysis", the readers have no idea which results?!

I would suggest that the authors rewrite this section. And first, explain the results and figures, then discuss whether or not their results agreed with other studies. Another problem is that figures don't have numbers. The captions with numbers are presented on a separate page and it's hard to follow Figures.

2- Looking at Figure.2 and supplementary Table S1, I see that in some countries (e.g., China, Venezuela, Ecuador) PSH is extended in almost all scenarios. For example, the result shows in China, with the model that uses Melting-0.5, the extend of the PSH increases by 2.73 during 2020-2060, which means climate change and melting scenarios are likely to be beneficial to many species in these countries. However, based on WWF reports, half of China's terrestrial vertebrates have vanished in the last 40 years and more decrease will be expected in the near future.

In this report, habitat loss, pollution and nature degradation by human activities and development are considered to be the most significant threats to biodiversity in China.

-Now, by reading the report developed by WWH, how do the authors interpret their results? For example, when their results show that China gaining more species, is there any other study, that can support these findings?

Same for the other 12 study areas.

Besides, climate change is not linear, rather is a complex pattern. For example, the probability of local climate extremes is predicted to be high in many parts of the world, including Latin America, northern Australia, Europe and a major part of Africa. Climate change velocity (magnitude and direction) is also increasing in many parts of the world and is patchy and depends on many factors. Considering all these, how the authors would interpret their results?

I strongly believe that the authors should be cautious about attributing causes to climate change and melting scenarios.

I wouldn't say doing more analysis, but these problems of attribution are inherent to the use of correlations for inferring causation. So, I would suggest a nice and strong discussion on this.

3- I think that would be useful if the authors add a map of changes in precipitation and temperature under the different scenarios. I suggest instead of figures S9-S10-10-S11, in supplementary material the authors add some maps of Spatio-temporal variation in precipitation and temperature under different scenarios. For example, authors can add a map of simple anomalies or shift in isotherm under different scenarios. This will be really informative and helpful to understand how different

variables have changed under different scenarios.

4- I'm just wondering how is the variation of TSS in the ensemble for each scenario. Which scenario had the highest TSS?

L-50: Reference number 9 is not relevant to this line.

L72-76: Awkwardly written please rephrase.

L-102: Irrelevant citation! None of these three studies cited by authors assessed a decline in species richness under the future scenarios of climate change. In general, there are many irrelevant citations in this manuscript, I would suggest to the authors that chose their citations carefully.

Reference number 9 for example, by Chris D.Thomas 2010, is about species range dynamics (not richness) under the ongoing climate change, not future scenarios.

Reference number 24 is also about the life history and spatial traits and not richness decline.

L-104: "The" should be added before temperature.

L-114: This line is repetition from the above lines.

L-117: Not really sure if the authors can draw the conclusion based on the data and analysis they used that "mountain ecosystems are more vulnerable". Reaching this conclusion needs more high-resolution data (microclimate) and more sophisticated approaches. For example, is already showed that the mountain's microclimate act as refuges for biodiversity (eg., Extinction risk from climate change is reduced by microclimatic buffering. AJ Suggitt, RJ Wilson, NJB Isaac, CM Beale, AG Auffret... - Nature Climate Change, 2018).

L-129: Not really sure what the authors mean by "The vast majority of the species modelled are distributed within or around biodiversity hotspots", this citation has nothing to do with biodiversity hotspots.

Reviewer #2 (Remarks to the Author):

Manuscript overview:

In the manuscript "Greenland's thaw pushes the biodiversity crisis", Ureta et al. aim to quantify the potential impacts of Greenland's glacial ice loss on the composition and distribution of tetrapods and vascular plants across 12 different countries in Australia, Asia, South America, and Africa. The authors utilize a suite of modeling techniques to evaluate the varying levels of change in biodiversity over the next century resulting from four different scenarios of sea level rise (0.5, 1, 1.5, and 3 meters). These four scenarios, caused by exacerbated glacial melting in Greenland, are added to baseline conditions predicted by the IPCC in a business-as-usual climate change scenario (RCP 8.5). The authors find a broad array of results (including range reductions, changes in species composition, and species richness) that may occur due to worsened ice loss in Greenland. They conclude a global collapse in biodiversity is possible in the next 10 – 40 years depending on the severity of ice loss.

Overall impression:

Ureta et al. set out to answer important and novel questions regarding the impact of sea level rise on commonly overlooked taxa throughout the world. Their modelling techniques appear to be rigorous and transparent, their figures are interesting and beautiful, and their topic is of suitable general appeal to justify publication in *Communications Biology*. However, I think they are hindered by the vast scope of their analysis. The vague generalizations that are found in their Results and Discussion section left me confused and craving more information. Several of their critically important decisions, such as which countries or taxa to include in their analysis, are presented without appropriate justification. The AMOC appears to be a central feature that connects melting and biodiversity, but the pathways of its impact are not clearly communicated. Their unique methodology is overshadowed by a puzzling, and often contradicting, presentation of their results. Additionally, many of their most interesting findings are left without an explanation. I also found myself questioning some results due to the potentially confounding effect of data reporting between countries, which is an issue that is highlighted by the authors several times in the manuscript.

I think there is tremendous potential in this study and I find the subject matter fascinating. However, I suggest significant alterations in how the material is presented in the main text. It is my understanding there is no word or page limit in *Communications Biology*. I generally recommend adding a significant level of detail in order to convince and influence scientists across different fields.

Specific comments:

1. Line 28: References 2 and 3 are specific to Arctic ice sheets surrounding Greenland. If the authors would like to use the word "polar" instead of "Arctic", I encourage at least one Antarctic reference.
2. Line 30: I think "unexplored" is a bit too strong of a word here given the efforts of several other groups. Instead, I would suggest something like, "poorly understood".
 - a. For example: Cauvy-Fraunié, S. and Dangles, O., 2019. A global synthesis of biodiversity responses to glacier retreat. *Nature ecology & evolution*, 3(12), pp.1675-1685.
 - b. And a response that sets the background nicely for your research: Stibal, M., Bradley, J.A., Edwards, A., Hotaling, S., Zawierucha, K., Rosvold, J., Lutz, S., Cameron, K.A., Mikucki, J.A., Kohler, T.J. and Šabacká, M., 2020. Glacial ecosystems are essential to understanding biodiversity responses to glacier retreat. *Nature ecology & evolution*, 4(5), pp.686-687.
3. Lines 34 – 35, 37 – 39, 132 – 133: I'm confused by these results, as median implies the singular midpoint of a distribution of values. Is 35 – 78% the 95% CI or standard error? Are these a range of medians for the 12 countries, or for different species? I suggest rewording, "median range loss" to more accurately define the results.
4. Line 48: When referencing the most important hotspots here, I think you should clarify you are focusing on only terrestrial (and freshwater I think) diversity.
5. Line 48: These countries should be named here or a table should be referenced. Additionally, why did you choose these 12 countries? Why did you decide to focus on 12 and not another quantity?
6. Line 49: Given the broad taxonomic scale of global biodiversity, it is unclear to me why the authors immediately direct their focus to these two subgroups (tetrapods and vascular plants). I understand taxonomic limitations are necessary for a study of this nature, but there should be an explanation behind this decision.
7. Lines 53 – 54: This statement requires references. I also suggest removing the reference to deforestation as it detracts from the climate change focus of the manuscript.
8. Lines 69 – 71: Given the relevance of Velasco et al. 2021 as a precursor to this manuscript, I recommend a little more information on their results and how they relate to this analysis. For

example, were many these amphibians also located in any of the 12 megadiverse countries? Furthermore, what are specific examples of how a weaker AMOC enhanced their decline?

9. Lines 85 – 87: The authors must elaborate as to why they chose to limit their analysis to these RCP scenarios. Why only variations of RCP 8.5 and not 2.6, 4.5, or 6? This information could be communicated elsewhere in the manuscript, but then there should be a reference directing readers (e.g., See Methods).

10. Lines 92 – 93: The authors should elaborate as to why they chose these temporal divisions. Again, the Methods section is fine but then there should be a reference.

11. Lines 101 – 102: This sentence could be better suited as Introduction material.

12. Lines 103 – 104: This is an important conclusion and I would like to see more evidence/discussion as to how your results suggest these variables (precipitation and temperature) are driving biodiversity.

13. Line 105: Instead of using “global warming” here I think you should stick with RCP 8.5.

14. Line 107: You mentioned the impacts of AMOC weakening in your Introduction section. However, it is unclear how your different melting scenarios impact the strength of AMOC.

15. Line 109: I think “excepting” might be a typo for “except in”.

16. Lines 109 – 110: This sentence seems to question the legitimacy of your own results? Why include Indonesia, India, and the Philippines in your analysis if the biological data is untrustworthy?

17. Line 113: Given results and discussion have been combined in this section, I think you should immediately explain why China differed from the rest of your 12 countries.

18. Line 120: Are these melting scenarios the same as those used in your analysis? If so, which level of severity is relevant (e.g., 0.5, 1, 3, etc.)? This reference (DeFrance et al. 2020) includes the influence of Antarctic melting.

19. Line 122 – 123: Again, the contrasting results from Chinese tetrapods need to be further explained.

20. Lines 123 – 127: These sentences (beginning with, “Differences in precipitation...”) need to be reworded. More or equally are two dramatically different results. What does less consistent imply statistically speaking? And why do you think it was different for tetrapods?

21. Lines 134 – 135: Again, why is China the outlier?

22. Line 136: I’m still bothered by the vagueness of “melting scenarios”. I think you should specify which of your scenarios you are referring to.

23. Lines 145 – 147: Is this a result of your analysis? If so, these species should be named somewhere.

24. Lines 149 – 151: Since you start this sentence with, “based on our models”, I find it odd that you reference a figure in another study (Velasco et al.) rather than something from this manuscript.

25. Line 155: What mitigation and protection strategies are you considering here? It appears your analysis is based entirely on RCP 8.5 and worse, which assumes business-as-usual for the most part?

26. Line 160: At this point in the manuscript, the specific pathways that Greenland's melting scenarios alter species richness in the 12 countries are still unclear to me.

27. Line 164: The result 31 – 83% is quite a large interval of possible outcomes. I would like to see these results broken down into more certain predictions. The addition of C.I.'s or standard error would also be helpful.

28. Line 169: Again, which melting scenarios are you referring to?

29. Line 182: What statistical test did you use to compare the risks of extinction between plants and animal species? The results should be presented here.

30. Lines 187 – 188: Needs to be rephrased. Range reduction is not ubiquitous (i.e., found everywhere) if there is variation between countries.

31. Lines 207 – 209: Again, the data limitations of countries used in this analysis seem to prevent accurate comparisons with other countries that are data-rich. An alternative strategy would be two separate analyses comparing a group of data-rich countries and a group of data-poor countries.

32. Line 211: This uncertainty should be quantified and reported in the main text or the Methods section should be referenced.

33. Line 214: Why is intermodal variability not considered? Reasoning should be referenced in Methods, or a reference to another study should be presented that further validates this approach.

34. Line 224 – 225: Why are there such a small number of ferns, gymnosperms, and lycophytes? Are they not as common? Are they not studied as frequently? As someone who studies vertebrates, I would appreciate more information regarding these limitations.

35. Line 235 – 236: This statement, indicating the influence of tetrapod diversity on pollination, needs a citation.

36. Line 244: It is still unclear to me how AMOC specifically impacts your 12 countries. Is this impact variable across different groups (e.g., vertebrates vs. vascular plants)? And how do the different melting scenarios impact the strength of AMOC?

37. Line 387 – 388: I believe this is the first mention that aquatic species were considered in this analysis. I think this should be discussed in the main text. Were the results of terrestrial vs. aquatic species dramatically different? If so, why?

38. Line 408 – 409: There appears to be a large temporal gap in your methodology. Where bioclimatic variables were considered for 1970 – 2000, but in the Introduction you imply scenarios should be considered from the baseline of 2020?

39. Lines 625 – 631: The y-axes of these figures are confusing to me. The description indicates positive values indicate range reductions, but by 2070 it appears that all taxonomic groups in all four melting scenarios reach -1, which means they all would experience range expansions? In fact, none of the taxonomic groups reach positive values (range reductions) in any melting scenario in any time period?

40. Lines 632 – 640: It is unclear what the different colors represent in figures a – c?

Reviewer #3 (Remarks to the Author):

Reviewer Assessment

Manuscript#: COMMSBIO-21-2012

Title: Greenland's thaw pushes the biodiversity crisis

Comments for Author

The manuscript entitled: "Greenland's thaw pushes the biodiversity crisis" examines globally the future change of distribution of a great number of plant species and tetrapods. The major claim is a reduction in species ranges and hotspots magnified by Greenland melting. The question they propose is interesting and it has been made a great effort compiling occurrence data of all the species and running the models. The authors have shared R code at Zenodo to make research reproducible and R plots are very nice.

About the novelty of the manuscript, consequences of Greenland's melting has already been assessed on amphibians (Velasco, J. A. et al 2021).

Main concerns:

My main concerns are related to these points: is this manuscript timely?, are the models produced for so many different taxa accurate?. These critics do not attempt to detract from the work done by the authors.

a) First, this manuscript is focused on the difference that Greenland's melting makes over projections made using the future RCP scenarios (pertaining to CMIP5) (in a similar way to Velasco et al 2021). However, CMIP6 climate models are already available and they contemplate greater Greenland ice sheet contribution compared to CMIP5 (see Hofer et al 2020). In my opinion, the rationale of the manuscript has become blurred by the availability of new future scenarios pertaining to the CMIP6 (Worldclim v2.1).

I understand COVID pandemic has delayed almost everyone scheduled research and using CMIP5 climate models would not make a big difference in some studies focused on other questions. But since this paper gives a lot of relevance to the influence of Greenland's melting, I think the manuscript would need to be updated using additionally (or exclusively) the CMIP6 climate models.

b) The methodology employed for ecological niche models seems standard and correct. However, I have a concern inherent to the large number of species and taxa used. It is regarding the selection of variables: was a previous selection of variables done? It seems only a correlation analysis was done. I find difficult to model the niche of more than 21,000 species pertaining to such different taxonomic groups by using the same variables for all of them... I mean, plants have very different requirements compared to mammals, for example. It is not clear to me how the most suitable variables to be included in the SDMs were selected for each species/taxonomic group.

c) How do you think the buffer size used to delimit M influences the results? I think it would be interesting to comment that the "no-dispersal" assumption affects the general results (see e.g. L131: "the geographic extent of potential species hotspots (PSHs) across countries decreases"). With this experimental design, the hypothesis of the species finding new suitable regions is very unlikely.

Minor changes:

L 28: What does "singular" mean here? Could you be more specific?

L 33: Please, reconsider the use of "high global warming" as it is expressed here

L 34. Why the median was chosen over the mean value in results?

L 407: WorldClim v.249 is not correctly referenced

L 458: the mean coefficient of variation – reflecting model uncertainty- needs further explanation and or reference.

L459-461: the two variables with the largest model contribution were used to inform on the results.

Please, make clear what method was used to calculate variables model contribution.

- Why do you think tetrapods models were less accurate in terms of ROC and TSS?

Figure 2: what does the first column = 1 show? Is it necessary to include it? (It is the same one in each graph)

Figure 4: the figure is very nice but I do not understand what the gradient of colors mean. I think it means nothing, but they are the same as for T1, T2... and you get distracted.

Supplementary (page 5) please add some parenthesis to the Species richness formula to add clarity

REFERENCES

Hofer, S., Lang, C., Amory, C., Kittel, C., Delhasse, A., Tedstone, A., & Fettweis, X. (2020). Greater Greenland Ice Sheet contribution to global sea level rise in CMIP6. *Nature communications*, 11(1), 1-11.

Velasco, J. A. et al. Synergistic impacts of global warming and thermohaline circulation collapse on amphibians. *Communications Biology* 4, 1-7 (2021).

Referee expertise:

Referee #1: Climate Change Ecology, Spatio-temporal Modeling, Species distribution

Referee #2: Oceanography, Marine Ecology, Polar Ecology, Fisheries

Referee #3: Biogeography, Biological Invasions, Climate Warming, Community Ecology

Reviewers' comments:

Reviewer #1 (Remarks to the Author):

#Reviewer comments

In this contribution, the authors measured the impact of climate change using a high-emission global warming scenario (RCP 8.5) and four melting scenarios, to model and predict the distribution of 221,146 species in 12 different countries.

The authors used their conclusion to highlight the importance of thermohaline circulation collapse in the future distribution of different taxonomic groups. They showed that melt-ing scenarios magnifies the decline in species geographical extend.

In summary, this contribution is an interesting work; and the results are nicely presented. Nonetheless, I have some major concerns and few very minor observations.

Major corrections

1- The result & Discussion section is poorly written. The result starts with reviewing the method and then jump to the discussion. For example, in line 102 which is the second paragraph of the result & discussion, when the authors mentioned “our results agree with previous analysis”, the readers have no idea which results?!

I would suggest that the authors rewrite this section. And first, explain the results and figures, then discuss whether or not their results agreed with other studies. Another problem is that figures don't have numbers. The captions with numbers are presented on a separate page and it's hard to follow Figures.

Answer: Thank you for your comment. We completely rephrased our results and discussion sections. We also incorporated the figure captures right after their corresponding figure to make the review process easier.

2- A) Looking at Figure.2 and supplementary Table S1, I see that in some countries (e.g., China, Venezuela, Ecuador) PSH is extended in almost all scenarios. For example, the result shows in China, with the model that uses Melting-0.5, the extend of the PSH increases by 2.73 during 2020-2060, which means climate change and melting scenarios are likely to be beneficial to many species in these countries. However, based on WWF reports,

half of China's terrestrial vertebrates have vanished in the last 40 years and more decrease will be expected in the near future. In this report, habitat loss, pollution and nature degradation by human activities and development are considered to be the most significant threats to biodiversity in China.

-Now, by reading the report developed by WWF, how do the authors interpret their results? For example, when their results show that China gaining more species, is there any other study, that can support these findings? Same for the other 12 study areas.

*Answer: We integrated the WWF report in our discussion and tried to clarify our results. As described in the WWF report, biodiversity is importantly decreasing given several anthropogenic factors such as land use change, pollution and anthropogenic climate change (WWF 2020). In our study, we evaluated the possible impact of climate change (worst case emission scenario) and an added effect coming from a catastrophic event such as an important amount of Greenland melting, without taking into account other anthropogenic factors that are in several cases a greater threat to biodiversity loss than climate change (WWF 2020). We are also aware that these anthropogenic factors can act synergically with climate change (Ureta et al. 2012, Martorell et al. 2015). Still, even when in our study only climate change was considered (with and without the added catastrophic event), current **Supplementary table S2** shows that all groups of species have very important losses in their suitable climatic conditions, going from about 30% loss to up to 100% loss across all scenarios evaluated. And even when **Supplementary table S2** shows slight increase in suitable climatic conditions under scenario RCP 2030 for countries such as China, Colombia and Venezuela; this suitability will be reduced in RCP 2050 and decreases in 2070 in all scenarios evaluated. This result is not uncommon, because a slight change in temperature or precipitation might be favorable for some species in some regions (Duan et al. 2016, Dyderski et al. 2018); but if a threshold is overpassed, than climatic conditions pass from suitable to unsuitable.*

*The hotspots increase showed by **figure 7** is in accordance to tables in the Supplementary material. As there is an increase in suitable climatic conditions in China and Venezuela, there are more areas with the potential of harboring a greater amount of species (potential hotspots). But as said before, this is only in terms of climatic suitability without taking into account other factors that might affect the persistence of species in a specific area, or that might even prevent species to get to their new suitable climatic conditions. Also, the climatic threshold is clear across time. Some change—as expected with RCP 8.5 T1—might be favorable, but once a threshold is overpassed a collapse is probable. From our results, it is difficult to determine the reason behind China being an exception from other countries, but we infer that it can be related to the fact that in comparison to the other countries evaluated: a) China has a higher seasonality than all other countries evaluated (**Supplementary figures S12–S22**), and consequently its endemic species might be more plastic biologically (Canale and Henry 2010), b) its annual*

temperature is also colder than most of the other countries evaluated and an increase in temperature might be beneficial for some species (Duan et al. 2016); and c) is a country with a large area with very high altitude (> 3000 masl) (Liu and Raven 2010). Species from high altitudes are regularly vulnerable to climate change because they do not have higher places to go to if conditions get warmer (Dirnböck et al. 2011, Urbani et al. 2017). In China, species do have geographic space to go to if conditions get warmer, expanding their distribution to higher altitudes (Figure 6), even when this expansion also represent species turnover (Supplementary figure 27). It is not uncommon that some temperature increase might become beneficial in terms of climatic suitability (Duan et al. 2016, and Figure 1), but time and melting scenarios increases the change in temperature and precipitation overpassing a threshold and that is why the potential hotspots (PHS) from figure 7 and Supplementary figures 27-32) decrease importantly. In general, even when melting scenarios do represent a weaker AMOC and consequently lower levels of warming, they also represent more dissimilarity in comparison to present-day climatic conditions leading to higher impacts than the once expected in RCP8.5 scenario.

2 B- Besides, climate change is not linear, rather is a complex pattern. For example, the probability of local climate extremes is predicted to be high in many parts of the world, including Latin America, northern Australia, Europe and a major part of Africa. Climate change velocity (magnitude and direction) is also increasing in many parts of the world and is patchy and depends on many factors. Considering all these, how the authors would interpret their results?

I strongly believe that the authors should be cautious about attributing causes to climate change and melting scenarios.

I wouldn't say doing more analysis, but these problems of attribution are inherent to the use of correlations for inferring causation. So, I would suggest a nice and strong discussion on this.

Answer: Thank you for your comment; we agree that attributions to climate change should mostly not be done. We will try to be more careful with our claims in our discussion. We are aware of the enormous uncertainty inherent to climate change simulations and consequently, studies analyzing potential impacts. An important evidence of this uncertainty is in part the main topic of this paper, because most climatic simulations have failed in including important climatic forces that could change simulations in a strong way such as Greenland melting. We will try to be more careful in our discussion; and include the potential harm of climate extremes that can cause severe damage. We also agree with the fact that local climate extremes can further impact biodiversity in several ways. In this study we only incorporated the synergistic effect of one catastrophic event (Greenland melting) with climate change; but we are aware of the fact that there are other extreme and catastrophic events that might further harm biodiversity in several ways such as fires and

hurricanes. We already incorporated this information in the text.

3- I think that would be useful if the authors add a map of changes in precipitation and temperature under the different scenarios. I suggest instead of figures S9-S10-10-S11, in supplementary material the authors add some maps of Spatio-temporal variation in precipitation and temperature under different scenarios. For example, authors can add a map of simple anomalies or shift in isotherm under different scenarios. This will be really informative and helpful to understand how different variables have changed under different scenarios.

*Answer: Thank you for your suggestion. We already incorporated the spatio-temporal variation maps complementing **Supplementary figures S20–S22 with S4-S19**. We decided not to delete **Supplementary figures S20–S22**, because even when maps helped us visualize in space and climatic scenarios differences in climatic variables, **Supplementary figures S20–S22** let us visualize in one single image all scenarios evaluated and compare them with the present-day scenario.*

4- I'm just wondering how is the variation of TSS in the ensemble for each scenario. Which scenario had the highest TSS?

*Answer: The code used helps obtaining the validation metrics (ROC and TSS) for each algorithm with which the modeling was carried out. Afterwards, we proposed an AUC threshold (>0.7 , coming from the ROC test) to create an ensemble map with the models that overpassed the threshold, giving higher weight to the models with greater performance (predictive power). The ensembled map was also evaluated with the validation metrics ROC and TSS and we presented these validation metrics in **Supplementary figure S1**. To present our data, ensemble projections were converted into binary maps by maximizing TSS. We would like to clarify that the validation metrics such as the ROC and the TSS were used for the ecological niche modeling algorithms under all scenarios and times, not to evaluate the climatic simulations. We already rephrased our methods and hope is clearer now.*

Minor comments

L-50: Reference number 9 is not relevant to this line.

Answer: Done.

L72-76: Awkwardly written please rephrase.

Answer: Done.

L-102: Irrelevant citation! None of these three studies cited by authors assessed a decline in species richness under the future scenarios of climate change. In general, there are many irrelevant citations in this manuscript, I would suggest to the authors that chose their citations carefully.

Reference number 9 for example, by Chris D. Thomas 2010, is about species range dynamics (not richness) under the ongoing climate change, not future scenarios.

Reference number 24 is also about the life history and spatial traits and not richness decline.

Answer: We tried to be more careful with our citations.

L-104: “The” should be added before temperature.

Answer: Done.

L-114: This line is repetition from the above lines.

Answer: Done.

L-117: Not really sure if the authors can draw the conclusion based on the data and analysis they used that “mountain ecosystems are more vulnerable”. Reaching this conclusion needs more high-resolution data (microclimate) and more sophisticated approaches. For example, is already showed that the mountain’s microclimate act as refuges for biodiversity (eg., Extinction risk from climate change is reduced by microclimatic buffering. AJ Suggitt, RJ Wilson, NJB Isaac, CM Beale, AG Auffret... - Nature Climate Change, 2018).

*Answer: Thank you for your comment. We agree with you that based on our results we cannot conclude that mountain ecosystems are more vulnerable. What we can say based on our results (**Supplementary figures S23-S26**) is that on average, regions with humid and temperature climates are predicted to be more vulnerable to climate change than regions with more seasonal and warmer conditions. We took out “mountain ecosystems” as our example. Our potentially highly diverse areas under current climatic scenario, RCP 8.5 and melting scenario coincide with geographic areas with higher altitudes in most countries evaluated. But these areas are expected to decrease with the exception of China. Please go to answer 2 for more detail about China. Consequently our results are in part in the same direction of those presented by Suggitt and coworkers (2018). Even when we did not evaluate climatic heterogeneity per se, we infer through our results that high altitude ecosystems will act as microrefugia for several species. Actually they are already acting as climatic refugia (Flousek et al. 2015). Still, the diversity that is already adapted to highland conditions, in a warmer scenario do not have another place to go to and are being reported*

to have negative impacts in their populations (Flousek et al. 2015). In other words, high altitude ecosystems are being an alternative for species that were living at lower-altitudes but not for the already high-altitude species. A species rechange in high altitudes is expected, but also species losses.

L-129: Not really sure what the authors mean by “The vast majority of the species modelled are distributed within or around biodiversity hotspots”, this citation has nothing to do with biodiversity hotspots.

Answer: We agree that it was confusing and decided to erase it.

Reviewer #2 (Remarks to the Author):

Manuscript overview:

In the manuscript “Greenland's thaw pushes the biodiversity crisis”, Ureta et al. aim to quantify the potential impacts of Greenland’s glacial ice loss on the composition and distribution of tetrapods and vascular plants across 12 different countries in Australia, Asia, South America, and Africa. The authors utilize a suite of modeling techniques to evaluate the varying levels of change in biodiversity over the next century resulting from four different scenarios of sea level rise (0.5, 1, 1.5, and 3 meters). These four scenarios, caused by exacerbated glacial melting in Greenland, are added to baseline conditions predicted by the IPCC in a business-as-usual climate change scenario (RCP 8.5). The authors find a broad array of results (including range reductions, changes in species composition, and species richness) that may occur due to worsened ice loss in Greenland. They conclude a global collapse in biodiversity is possible in the next 10–40 years depending on the severity of ice loss.

Overall impression:

Ureta et al. set out to answer important and novel questions regarding the impact of sea level rise on commonly overlooked taxa throughout the world. Their modelling techniques appear to be rigorous and transparent, their figures are interesting and beautiful, and their topic is of suitable general appeal to justify publication in *Communications Biology*. However, I think they are hindered by the vast scope of their analysis. The vague generalizations that are found in their Results and Discussion section left me confused and craving more information. Several of their critically important decisions, such as which countries or taxa to include in their analysis, are presented without appropriate justification. The AMOC appears to be a central feature that connects melting and biodiversity, but the pathways of its impact are not clearly communicated. Their unique methodology is overshadowed by a puzzling, and often contradicting, presentation of their results. Additionally, many of their most interesting findings are left without an explanation. I also found myself questioning some results due to the potentially confounding effect of data reporting between countries, which is an issue that is highlighted by the authors several times in the manuscript.

I think there is tremendous potential in this study and I find the subject matter fascinating. However, I suggest significant alterations in how the material is presented in the main text. It is my understanding there is no word or page limit in *Communications Biology*. I generally recommend adding a significant level of detail in order to convince and influence scientists across different fields.

Answer: Thank you for your major comments. We tried to rephrase our results and discussion section to clarify and addressed most of your concerns. We also attended each particular comment from your “specific comments” section.

Major and minor comments

1. Line 28: References 2 and 3 are specific to Arctic ice sheets surrounding Greenland. If the authors would like to use the word “polar” instead of “Arctic”, I encourage at least one Antarctic reference.

Answer: We changed the word “polar” for “Arctic”.

2. Line 30: I think “unexplored” is a bit too strong of a word here given the efforts of several other groups. Instead, I would suggest something like, “poorly understood”.

a. For example:

Cauvy-Fraunié, S. and Dangles, O., 2019. A global synthesis of biodiversity responses to glacier retreat. *Nature ecology & evolution*, 3(12), pp.1675-1685.

b. And a response that sets the background nicely for your research:

Stibal, M., Bradley, J.A., Edwards, A., Hotaling, S., Zawierucha, K., Rosvold, J., Lutz, S., Cameron, K.A., Mikucki, J.A., Kohler, T.J. and Šabacká, M., 2020. Glacial ecosystems are essential to understanding biodiversity responses to glacier retreat. *Nature ecology & evolution*, 4(5), pp.686-687.

Answer: We rephrased our abstract and erased the word “unexplored”. Thank you for the suggested references.

3. Lines 34 – 35, 37 – 39, 132 – 133: I’m confused by these results, as median implies the singular midpoint of a distribution of values. Is 35 – 78% the 95% CI or standard error? Are these a range of medians for the 12 countries, or for different species? I suggest rewording, “median range loss” to more accurately define the results.

Answer: These values are ranges for the medians estimated across groups of species under different scenarios. We decided to estimate a single median for all species for the main text, and refer to Supplementary table 2–4 for medians calculated by country or group. We already clarified in our text that those ranges correspond to the range loss of species from all groups and countries evaluated under all scenarios.

4. Line 48: When referencing the most important hotspots here, I think you should clarify you are focusing on only terrestrial (and freshwater I think) diversity.

Answer: Done.

5. Line 48: These countries should be named here or a table should be referenced. Additionally, why did you choose these 12 countries? Why did you decide to focus on 12 and not another quantity?

Answer: We already named in our manuscript the megadiverse countries that we took into account and why.

6. Line 49: Given the broad taxonomic scale of global biodiversity, it is unclear to me why the authors immediately direct their focus to these two subgroups (tetrapods and vascular plants). I understand taxonomic limitations are necessary for a study of this nature, but there should be an explanation behind this decision.

Answer: Thank you for your comment. It was not explicit why we decided to work with those groups. We tried to clarify our decisions in the main text including that minimum geographic information is needed to carry out an analysis as the presented in our work. We also described further taxonomic complications that exist in other groups (Mueller and Schmit 2007, Naranjo-Ortiz and Gabaldón 2019).

7. Lines 53 – 54: This statement requires references. I also suggest removing the reference to deforestation as it detracts from the climate change focus of the manuscript.

Answer: We already added the corresponding references and removed the reference to deforestation.

8. Lines 69 – 71: Given the relevance of Velasco et al. 2021 as a precursor to this manuscript, I recommend a little more information on their results and how they relate to this analysis. For example, were many these amphibians also located in any of the 12 megadiverse countries? Furthermore, what are specific examples of how a weaker AMOC enhanced their decline?

Answer: Thank you for the comment. We tried to give more information on how our work relates to what was found by Velasco's results. We also clarified that amphibians in Velasco's work were evaluated across the globe and consequently he incorporated all the amphibians from megadiverse countries. A weaker AMOC had consequences in the climatic layers with which the ecological niche modeling was carried out to identify climatic suitability across the world. In comparison to climatic layers from RCP 8.5, decreases in climatic suitability under melting scenarios were more evident. That is the way we also evaluated potential impacts of a weaker AMOC. Finally, we added that in general, even when a weaker AMOC is expected to have a heterogeneous effect spatially, a

slowdown in climate change temperature is projected but also further alterations of regional climates that further increment the pressure on biodiversity. Please find this information in our reviewed version.

9. Lines 85 – 87: The authors must elaborate as to why they chose to limit their analysis to these RCP scenarios. Why only variations of RCP 8.5 and not 2.6, 4.5, or 6? This information could be communicated elsewhere in the manuscript, but then there should be a reference directing readers (e.g., See Methods).

Answer: We included information about why we used RCP 8.5 in our method section but also in our introduction. We argued that, melting scenario would have been more difficult to be supported by lower emission scenarios. However; that does not preclude a large melting under lower emission scenarios. Furthermore, we are using a well-designed opportunity experiments from DeFrance et al. (2017) and wanted to stick with their choice: RCP 8.5.

10. Lines 92 – 93: The authors should elaborate as the why they chose these temporal divisions. Again, the Methods section is fine but then there should be a reference.

Answer: We explained why we chose our temporal divisions in our introduction (briefly) and in our method section: “Time horizons evaluated aim to represent short, medium and long terms to help decision makers order conservation priorities.”

11. Lines 101 – 102: This sentence could be better suited as Introduction material.

Answer: We changed the sentence.

12. Lines 103 – 104: This is an important conclusion and I would like to see more evidence/discussion as to how your results suggest these variables (precipitation and temperature) are driving biodiversity.***Melting has a large added impact***

*Answer: We already included extra figures in our Supplementary material complementing **Supplementary figures S20–S22**, with **Supplementary figures 4-19**, which represent changes between present-day scenario and future scenarios. As it is possible to visualize, melting scenarios further increase changes in precipitation and temperature. The greater the changes in precipitation and temperature the further away are the suitable conditions of present day species. Even when melting scenarios represent a weaker AMOC and consequently lower warming levels, they also represent more dissimilarity to current climatic conditions.*

13. Line 105: Instead of using “global warming” here I think you should stick with RCP 8.5.

Answer: Done.

14. Line 107: You mentioned the impacts of AMOC weakening in your Introduction section. However, it is unclear how your different melting scenarios impact the strength of AMOC.

Answer: Thank you for your comment. We now explained in our main text that an increment of water in the ocean results in a weaker AMOC which consequences are different alterations in the global climate system and in regional climates. The greater the amount of water that is discharged into the ocean (scenarios: 0.5, 1, 1.5 and 3 masl), the weaker the AMOC. Also, in general a weaker AMOC is expected to decelerate climate change warming. However; melting scenarios increases dissimilarities with present-day climatic conditions in comparison to RCP 8.5. Even when RCP 8.5 might represent in general, warmer conditions, an increase in climatic dissimilarity by melting scenarios or time horizons have greater impacts on species climatic suitability. We already tried to clarify the relationship between the AMOC and our different scenarios. The physics behind how Greenland melting affects the AMOC is reported in Swingedouw et al. (2015). We will not add much more detail in our introduction because we think that deepening about the AMOC physics in our introduction section might mislead the reader.

15. Line 109: I think “excepting” might be a typo for “except in”.

Answer: The wording changed.

16. Lines 109 – 110: This sentence seems to question the legitimacy of your own results? Why include Indonesia, India, and the Philippines in your analysis if the biological data is untrustworthy?

*Answer: Thank for you comment. We recognize that there are geographic areas that present increases in climatic suitability for some species (current **Figure 6 and Supplementary Figure 2**), but that does not mean that there is a general tendency for species increase of climatic suitability. As shown in current **Figure 2**, the general tendency in all groups and countries evaluated across the time horizons used is a decrease in suitability range (potential distribution range). And as explained above, even when some climatic changes might become beneficial for some species in some areas as shown by our results and in other studies (i.e., figure 1: China, Venezuela and Colombia under RCP 8.5 T1, Duan et al. 2016, Dyderski et al. 2018), there is a threshold that if gets overpassed,*

than strong reductions in climatic suitability are expected (Velasco et al. 2021). We rephrased our results in order to avoid confusion. It is true that Indonesia, India and Philippines might have scarce data for some groups evaluated, but they still show similar tendencies to those countries with a higher number of species modeled (i.e., Mexico, Australia and Brazil): stronger reductions with melting scenarios and across time.

17. Line 113: Given results and discussion have been combined in this section, I think you should immediately explain why China differed from the rest of your 12 countries.

*Answer: Thank you for your comment, we already explained with detail why China is an exception in terms of climatic suitability and potential hotspots (PHS). Why China in particular resulted to be a country in which climatic suitability might increase for some species is hard to determine with our data and results, but we infer that: a) China has a higher seasonality than all other countries evaluated (**Supplementary figures S12–S22**), and consequently its endemic species might be more plastic biologically (Canale and Henry 2010), b) its annual temperature is also colder than most of the other countries evaluated and an increase in temperature might be beneficial for some species (Duan et al. 2016); and c) is a country with a large area with very high altitude (> 3000 masl) (Liu and Raven 2010). Species from high altitudes are regularly vulnerable to climate change because they do not have higher places to go to if conditions get warmer (Dirnböck et al. 2011, Urbani et al. 2017). In China, species do have geographic space to go to if conditions get warmer, expanding their distribution to higher altitudes (**Figure 6**), even when this expansion also represent species turnover (**Supplementary figure 27**).*

*We also incorporated new **Supplementary figures (S27–S32)** that helped us visualize that the increase in PHS strongly depends on the threshold used. In other studies, different scenarios might present some climatic changes that increase the climatic suitability for some species in China (Duan et al. 2016). But there is a climatic threshold that if it is overpassed, than climatic suitability can decrease importantly (Velasco et al. 2021). There are climatic ranges in which species can exist and persist through time. This climatic range is part of the species' ecological niche, but it is a range with climatic limits.*

18. Line 120: Are these melting scenarios the same as those used in your analysis? If so, which level of severity is relevant (e.g., 0.5, 1, 3, etc.)? This reference (Defrance et al. 2020) includes the influence of Antarctic melting.

Answer: We already clarified that those melting scenarios we are referring to are the same once with which we carried out our analyses. We also mentioned that all of them are severe. It seems that the 0.5 melting scenario is a tipping point. We also erased the reference from Defrance 2020.

19. Line 122 – 123: Again, the contrasting results from Chinese tetrapods need to be further explained.

*Answer: As it is explained in answer 17, with our results it is not possible to conclude why China's biodiversity is the exception in several results, but we infer that: a) China has a higher seasonality than all other countries evaluated (**Supplementary figures S12–S22**), and consequently its endemic species might be more plastic biologically (Canale and Henry 2010), b) its annual temperature is also colder than most of the other countries evaluated and an increase in temperature might be beneficial for some species (Duan et al. 2016); and c) is a country with a large area with very high altitude (> 3000 masl) (Liu and Raven 2010). Species from high altitudes are regularly vulnerable to climate change because they do not have higher places to go to if conditions get warmer (Dirnböck et al. 2011, Urbani et al. 2017). In China, species do have geographic space to go to if conditions get warmer, expanding their distribution to higher altitudes (**Figure 6**), even when this expansion also represent species turnover (**Supplementary figure 27**).*

20. Lines 123 – 127: These sentences (beginning with, “Differences in precipitation...”) need to be reworded. More or equally are two dramatically different results. What does less consistent imply statistically speaking? And why do you think it was different for tetrapods?

*Answer: We reworded the entire Results and Discussion section. This section now can be read as follows: “In general, our data suggest that regions with more temperate climates have greater losses in species' climatic suitability than regions with more seasonal and warmer climates (**Supplementary figure 23–26**). These results agree with the predicted reduction in Equatorial and warm temperate climates predicted under climate change scenarios (that is, climates A and C in the Köppen classification) (Defrance et al. 2020).”*

21. Lines 134 – 135: Again, why is China the outlier?

Answer: Please see our response in line 17. Our manuscript has been rephrased.

22. Line 136: I'm still bothered by the vagueness of “melting scenarios”. I think you should specify which of your scenarios you are referring to.

Answer: We already tried to specify which melting scenarios we are referring to.

23. Lines 145 – 147: Is this a result of your analysis? If so, these species should be named somewhere.

Answer: We were referring to endemic species harbored at biodiversity hotspots in megadiverse countries. We rephrased the sentence: “Not surprisingly and given our selection of endemic species, the PSH coincide with globally important biodiversity hotspots (Myers et al. 2000), which harbor an important percentage of the endemic and threaten species of the world.”

24. Lines 149 – 151: Since you start this sentence with, “based on our models”, I find it odd that you reference a figure in another study (Velasco et al.) rather than something from this manuscript.

Answer: You are right. We are now making reference to our own figures that show dramatic declines and alteration of biodiversity across megadiverse countries within a relatively short period of time.

25. Line 155: What mitigation and protection strategies are you considering here? It appears your analysis is based entirely on RCP 8.5 and worse, which assumes business-as-usual for the most part?

Answer: Thank you for your comment. You are right, we did not evaluate protection. We completely rephrased our sentence.

26. Line 160: At this point in the manuscript, the specific pathways that Greenland’s melting scenarios alter species richness in the 12 countries are still unclear to me.

Answer: Greenland melting scenarios further change bioclimatic variables from current conditions in comparison to RCP 8.5; decreasing the climatic suitability for several species. Given that species rich areas are the ones that have the potential of harboring more species, if climatic suitability reduces for several species, species rich areas will also decrease. We rephrased our entire results and discussion section and we hope it is clearer now.

27. Line 164: The result 31–83% is quite a large interval of possible outcomes. I would like to see these results broken down into more certain predictions. The addition of C.I.’s or standard error would also be helpful.

Answer: We agree that it is a large interval. The range represents the medians estimated across all groups of species evaluated and that is the reason why the interval is large. The

*estimates are broken down in the **Supplementary table 2**. We changed our wording to clarify. Also, **Figure 4** shows the entire distribution of species range sizes per group under RCP 8.5 and Melting 0.5 (tipping point) at the three time horizons; consequently, it is possible to visualize the CI.*

28. Line 169: Again, which melting scenarios are you referring to?

Answer: We incorporated to which melting scenario we are referring to.

29. Line 182: What statistical test did you use to compare the risks of extinction between plants and animal species? The results should be presented here.

Answer: We did not use any kind of statistical analysis. We claimed that plants are at higher risk than tetrapods by looking at their range loss. However; we changed our claim by writing down that on average plants are expected to be more vulnerable than animals.

30. Lines 187 – 188: Needs to be rephrased. Range reduction is not ubiquitous (i.e., found everywhere) if there is variation between countries.

Answer: We already rephrased.

31. Lines 207 – 209: Again, the data limitations of countries used in this analysis seem to prevent accurate comparisons with other countries that are data-rich. An alternative strategy would be two separate analyses comparing a group of data-rich countries and a group of data-poor countries.

Answer: We think separating countries in two groups will not give us different insights or patterns. Mexico, Australia and Brazil are countries with an important amount of data and their results under RCP 8.5 and melting scenarios is that strong losses in climatic suitability for most of the species evaluated are expected. In other words, the big picture would still be very similar.

32. Line 211: This uncertainty should be quantified and reported in the main text or the Methods section should be referenced.

*Answer: We quantified the uncertainty coming from different ecological niche algorithms by calculating the variation coefficient: “The coefficient of variation of our models (reflecting the degree of agreement/disagreement in predictions across algorithms) is in the range of 0.054–0.081, reflecting robust modeling results, at least under the modelling conditions we used (**Supplementary table 1**)”.*

33. Line 214: Why is intermodal variability not considered? Reasoning should be referenced in Methods, or a reference to another study should be presented that further validates this approach.

Answer: The variability among circulation models is not considered due to the fact that the melting models we used were constructed under the IPSL-CM5-LR GCM and are not available for any other GCMs. We added this explanation to the methods. “We acknowledge that using a single GCM does not allow us to estimate inter-GCM variability in the resulting distribution models. However; the melting scenarios do only exist for IPSL-CM5-LR GCM”.

34. Line 224 – 225: Why are there such a small number of ferns, gymnosperms, and lycophytes? Are they not as common? Are they not studied as frequently? As someone who studies vertebrates, I would appreciate more information regarding these limitations.

Answer: The causes for the lack of information about some groups of plants are multiple. The main reason behind these smaller numbers is the overall diversity of these groups: ~1,000 extant gymnosperms, ~1,000 extant lycophytes and 10,000 extant ferns. These numbers are substantially smaller than the estimated ~300,000 flowering plants (Stevens 2001, Joppa et al. 2011). In addition, in the case of ferns, these are believed to have lower levels of endemism owing to their biological characteristics, such as wind-dispersed spores. Gymnosperms are more diverse in northern latitudes and not in the, mostly tropical, megadiverse countries (Fraginière et al. 2015). This can partly explain the low number of endemics. Of course, there might be other data biases for these groups, which in many cases are less known than their flowering counterparts. We added the above explanation in the main text.

35. Line 235 – 236: This statement, indicating the influence of tetrapod diversity on pollination, needs a citation.

Answer: We incorporated scientific references to our statement.

36. Line 244: It is still unclear to me how AMOC specifically impacts your 12 countries. Is this impact variable across different groups (e.g., vertebrates vs. vascular plants)? And how do the different melting scenarios impact the strength of AMOC?

Answer: The AMOC experiments were superimposed to the RCP 8.5 scenario adding 0.11, 0.22, 0.34, and 0.68 Sv (1 Sv = 10^6 m³/s), that correspond to different increases in sea level

rise (0.5, 1, 1.5 and 3 masl) coming from a freshwater release from 2020 to 2070 (Anthoff et al. 2016). What we evaluated in this study is how these experiments lead to different changes in the climatic system by an AMOC weakening; and how these changes affect species climatic suitability across the countries evaluated. In general, the greater the amount of water release the weaker is the AMOC and a deceleration of global warming is expected (Anthoff et al. 2016). Changes in the climatic system across the countries evaluated are heterogeneous as it can be seen in **Supplementary figures 4–22**. But the general response of all groups evaluated is a strong reduction in RCP 8.5 that increases deeply with melting scenarios (**Figure 2**); because even when they might lower down some warming, their climatic dissimilarity to current climatic conditions is greater than RCP 8.5. These greater changes in the climatic systems result in an important increase of endemic species losing their entire climatic suitability in their countries. We tried to incorporate this information in the manuscript to clarify.

37. Line 387 – 388: I believe this is the first mention that aquatic species were considered in this analysis. I think this should be discussed in the main text. Were the results of terrestrial vs. aquatic species dramatically different? If so, why?

Answer: The vast majority of the species evaluated were terrestrial and only few of them (i.e., 8 mammals) depend on freshwater to complete their life cycle. In the earlier version of the manuscript we made such distinction given that some data bases from which we obtained the records to carry out our analysis, classified some species groups that way. However; it was not our aim to evaluate differences between complete terrestrial and semi-aquatic species. We decided to erase such distinction to avoid confusion. Still, we are aware of the fact that these species will have further restrictions to get to their new suitable climatic conditions as they depend on having a water body close to them.

38. Line 408 – 409: There appears to be a large temporal gap in your methodology. Where bioclimatic variables were considered for 1970–2000, but in the Introduction you imply scenarios should be considered from the baseline of 2020?

Answer: Current climatic conditions are represented by the following decades 1970–2000, in which changes by climate change equals zero. From this time horizon bioclimatic variables start to vary by climate change. We chose the other three time horizons because they help having projections on the short, medium and long term helping decision makers order conservation priorities. The simulations coming from Greenland melting start in 2020. Greenland melting scenarios are an additional effect to climate change. We already tried to clarify this information in our method section “Climate data”.

39. Lines 625 – 631: The y-axes of these figures are confusing to me. The description indicates positive values indicate range reductions, but by 2070 it appears that all

taxonomic groups in all four melting scenarios reach -1, which means they all would experience range expansions? In fact, none of the taxonomic groups reach positive values (range reductions) in any melting scenario in any time period?

Answer: Thank you for your observation. It was wrongly described. We already changed it. Negative values indicate ranges reductions and positive values indicate range expansions.

40. Lines 632 – 640: It is unclear what the different colors represent in figures a – c?

Answer: Different colors represent: a) species that gain suitable climatic conditions, b) species that have a moderate loss, c) severe loss, d) extreme loss or e) complete loss in the different scenarios evaluated.

Reviewer #3 (Remarks to the Author):

Reviewer Assessment

Manuscript#: COMMSBIO-21-2012

Title: Greenland's thaw pushes the biodiversity crisis

Comments for Author

The manuscript entitled: “Greenland's thaw pushes the biodiversity crisis” examines globally the future change of distribution of a great number of plant species and tetrapods. The major claim is a reduction in species ranges and hotspots magnified by Greenland melting. The question they propose is interesting and it has been made a great effort compiling occurrence data of all the species and running the models. The authors have shared R code at Zenodo to make research reproducible and R plots are very nice.

About the novelty of the manuscript, consequences of Greenland’s melting has already been assessed on amphibians (Velasco, J. A. et al 2021).

Major comments

My main concerns are related to these points: is this manuscript timely?, are the models produced for so many different taxa accurate?. These critics do not attempt to detract from the work done by the authors.

1- First, this manuscript is focused on the difference that Greenland's melting makes over projections made using the future RCP scenarios (pertaining to CMIP5) (in a similar way to Velasco et al 2021). However, CMIP6 climate models are already available and they contemplate greater Greenland ice sheet contribution compared to CMIP5 (see Hofer et al 2020). In my opinion, the rationale of the manuscript has become blurred by the availability of new future scenarios pertaining to the CIMP6 (Worldclim v2.1).

I understand COVID pandemic has delayed almost everyone scheduled research and using CMIP5 climate models would not make a big difference in some studies focused on other questions. But since this paper gives a lot of relevance to the influence of Greenlands melting, I think the manuscript would need to be updated using additionally (or exclusively) the CMIP6 climate models.

Answer: We agree with the reviewer that CMIP6 models are now available, and it might be certainly interesting to use these new models. Nevertheless, to our knowledge, none of them is using large amount of Greenland melting from extreme melting scenario as highlighted

in IPCC AR6 SPM Fig. 8.d, with contribution of ice sheet to sea-level reaching up to 3 m in a new worst-case scenario. The effect of such large melting still needs to be evaluated.

Furthermore, by reading the comment of the reviewer, we believe there is a misunderstanding here. The study of Hofer et al (2020) is not stating that CMIP6 simulations are including more Greenland melting. In fact, as in CMIP5, CMIP6 models are neglecting Greenland melting because this is very difficult to couple an ice-sheet model with climate models (there are several difficulties related to spatial resolution in CMIP6 models which are too coarse to couple with km-scale ice sheet models, biases in mean climatology of CMIP6 models, large memory of ice sheets, etc.). This is why Greenland melting is usually estimated using Regional Climate Models (RCM), and this is actually what is done in Hofer et al. (2020). Thus the Greenland melting estimates from this study is coming from one way coupling: CMIP6 models drive RCM, which then allow to estimate surface mass balance of Greenland, and deduce its melting. The melting is then not incorporated in CMIP6 projections, which have already been runned. This deficiency, present from CMIP5, is still there in CMIP6 models. The fact that those CMIP6 models drive larger Greenland melting actually further supports the rationale of our study: the potential of large Greenland melting should be evaluated, and since both CMIP5 and CMIP6 do not account for such large melting, this has not been done up to now.

While there remains uncertainty on the exact effect of such a melting on the AMOC, we argue here that this is not the topic of this study, which is focusing on a specific scenario, namely: what are the consequences of an AMOC weakening on biodiversity? This is a topic that is not sufficiently analyzed in the literature, as highlighted in IPCC SROCC (2019) in its Chapter 6.7 and Summary for Policymakers (SPM). Furthermore, the newly released IPCC AR6 report states in its SPM (2021) that: “There is medium confidence that there will not be an (AMOC) abrupt collapse before 2100. If such a collapse were to occur, it would very likely cause abrupt shifts in regional weather patterns and water cycle, such as a southward shift in the tropical rain belt, weakening of the African and Asian monsoons and strengthening of Southern Hemisphere monsoons, and drying in Europe.”

The collapse of the AMOC during this century is therefore not impossible, and even though it might be a low probability scenario, its potential high impacts need to be properly assessed (e.g. Sutton 2018). We believe that our study is clearly answering this type of uncertainty. We agree that it would have been ideal to use a CMIP6 climate model instead of this CMIP5 one. Nevertheless, we believe that this might not change the main conclusions obtained here, because fingerprints of an AMOC collapse does not seem to be very different from CMIP5 to CMIP6 (e.g., Jackson et al. 2015) who used a higher resolution climate model from CMIP6 generation and found similar fingerprints in terms of climatic impacts. This might be explained by the fact that those two families of scenarios are still relatively close, and that the main climatic effects of the AMOC is already well-represented climate dynamics in CMIP5.

Furthermore, a recent study show that the different rates of AMOC changes in CMIP6 is explaining much of the spread (and therefore uncertainty) in CMIP6 climate projections for the North Atlantic (Bellomo et al. 2021), clearly highlighting that this

process is crucial for climate projections, but the fate of the AMOC remains largely unknown quantitatively, due to complex non-linear dynamics, so that its potential impacts beyond climate should be assessed, as is done here.

Running similar experimental design with a CMIP6 model will deserve considerable computing time and effort, and might constitute a new project. Thus, we believe that this is beyond the scope of the present study, also given the arguments highlighted before. We have added a discussion of this shortcoming and about the arguments that makes us think this is not a major shortcoming of the present study, but might be certainly worth assessing in future studies.

2- The methodology employed for ecological niche models seems standard and correct. However, I have a concern inherent to the large number of species and taxa used. It is regarding the selection of variables: was a previous selection of variables done? It seems only a correlation analysis was done. I find difficult to model the niche of more than 21,000 species pertaining to such different taxonomic groups by using the same variables for all of them... I mean, plants have very different requirements compared to mammals, for example. It is not clear to me how the most suitable variables to be included in the SDMs were selected for each species/taxonomic group.

Answer: Thank you for your comment. We agree that same variables cannot be used for species from several groups, but even not for species from the same group. That is why, for each evaluated species we selected variables that presented less correlation between them ($R > 0.8$) in their specific M (geographic area in which the model was calibrated and projected). The calibration M was obtained from the intersection between a 4° buffer around species occurrences and terrestrial ecoregions. The projected M was a 2° buffer around the calibration area. We assumed climatic niche conservatism across time; and inside the projected M we also assumed full dispersal. Consequently, inside the projected M , the evaluated species can win or lose suitable climatic conditions. The variables and the number of variables to carry out the modeling were evaluated for each species individually. We already tried to clarify this information in our method section.

3- How do you think the buffer size used to delimit M influences the results? I think it would be interesting to comment that the “no-dispersal” assumption affects the general results (see e.g. L131: “the geographic extent of potential species hotspots (PSHs) across countries decreases”). With this experimental design, the hypothesis of the species finding new suitable regions is very unlikely.

Answer: With our experimental design species are able to find new suitable conditions inside their projected M . The calibration M was created based on the intersection between the ecoregions where the species has been registered and a 4° buffer, but then the

projected M has an additional 2° buffer around the calibration area. We are trying to take into account a more realistic dispersal scenario given the velocity with which climatic changes are happening. We do not think that extending our M to, for example, an entire continent would give helpful information. There are also geographic and ecological barriers, reason why we used ecoregions to limit our M. We already incorporated this information in our manuscript. Please see answer 2.

Minor comments

L 28: What does “singular” mean here? Could you be more specific?

Answer: The abstract was rephrased.

L 33: Please, reconsider the use of “high global warming” as it is expressed here

Answer: The abstract was rephrased.

L 34. Why the median was chosen over the mean value in results?

Answer: The median is more representative of the central tendency when we have very small or very large values.

L 407: WorldClim v.249 is not correctly referenced

Answer: We changed the reference to Fick, S.E. and R.J. Hijmans, 2017. WorldClim 2: new 1km spatial resolution climate surfaces for global land areas. International Journal of Climatology 37 (12): 4302-4315.

L 458: the mean coefficient of variation – reflecting model uncertainty- needs further explanation and or reference.

Answer: We already added in our main text that the coefficient of variation of our models ranges between 0.054–0.081, representing high consensus and low uncertainty between the algorithms used.

L459-461: the two variables with the largest model contribution were used to inform on the results. Please, make clear what method was used to calculate variables model contribution.

Answer: We already tried to clarify in our manuscript that the BIOMOD platform has several algorithms (up to nine) that use all selected variables (that were selected through a correlation analysis). The algorithms choose several variables combinations until it has the combination that better explains the distribution of the species. Those variables that have

more explicative power in the distribution of the species are the ones contributing more to the model.

- Why do you think tetrapods models were less accurate in terms of ROC and TSS?

Answer: Both groups of organisms had good performance in their modeling in terms of predictive power (Supplementary figure 34 and 35). However; we think plants performed better because it was possible to model with records. Instead, animals' distributions had to be approached by IUCN polygons. We incorporated this information in our main text.

Figure 2: what does the first column = 1 show? Is it necessary to include it? (It is the same one in each graph)

Answer: This figure is now Figure 7. Is the present-day result and we already clarified it in the text. Present-day scenario is the same for the RCP and the melting scenarios and we kept this column because we think it helps to quickly identify differences.

Figure 4: the figure is very nice but I do not understand what the gradient of colors mean. I think it means nothing, but they are the same as for T1, T2... and you get distracted.

Answer: The past Figure 4 is now Figure 3. Proportion of species of vascular plants and tetrapods with complete range loss across twelve megadiverse countries. The colors represent groups of species. In one single group of species there is no color gradient but different circle sizes. We joined each figure legend with its corresponding figure to avoid confusion. I hope it is clearer now.

Supplementary (page 5) please add some parenthesis to the Species richness formula to add clarity.

Answer: Done.

References

- Anthoff, D., F. Estrada, and R. S. Tol. 2016. Shutting down the thermohaline circulation. *American Economic Review* **106**:602-606.
- Bellomo, K., M. Angeloni, S. Corti, and J. von Hardenberg. 2021. Future climate change shaped by inter-model differences in Atlantic meridional overturning circulation response. *Nature Communications* **12**:1-10.
- Canale, C. I. and P.-Y. Henry. 2010. Adaptive phenotypic plasticity and resilience of vertebrates to increasing climatic unpredictability. *Climate research* **43**:135-147.
- Collins M., M. Sutherland, L. Bouwer, S. M. Cheong, T. Frölicher, H. Jacot Des Combes, M. Koll Roxy, I. Losada, K. McInnes, B. Ratter, E. Rivera-Arriaga, R. D. Susanto, D. Swingedouw, and L. Tibig. 2019. Extremes, Abrupt Changes and Managing Risk. *in* H. O. Pörtner, D. C. Roberts, V. Masson-Delmotte, M. T. Zhai, P., E. Poloczanska, K. Mintenbeck, A. Alegría, M. Nicolai, A. Okem, J. Petzold, B. Rama, and N. M. Weyer, editors. IPCC Special Report on the Ocean and Cryosphere in a Changing Climate
- Defrance, D., T. Catry, A. Rajaud, N. Dessay, and B. Sultan. 2020. Impacts of Greenland and Antarctic Ice Sheet melt on future Köppen climate zone changes simulated by an atmospheric and oceanic general circulation model. *Applied Geography* **119**:102216.
- Defrance, D., G. Ramstein, S. Charbit, M. Vrac, A. M. Famien, B. Sultan, D. Swingedouw, C. Dumas, F. Gemenne, and J. Alvarez-Solas. 2017. Consequences of rapid ice sheet melting on the Sahelian population vulnerability. *Proceedings of the national Academy of Sciences* **114**:6533-6538.
- Dirnböck, T., F. Essl, and W. Rabitsch. 2011. Disproportional risk for habitat loss of high-altitude endemic species under climate change. *Global Change Biology* **17**:990-996.
- Duan, R.-Y., X.-Q. Kong, M.-Y. Huang, S. Varela, and X. Ji. 2016. The potential effects of climate change on amphibian distribution, range fragmentation and turnover in China. *PeerJ* **4**:e2185.
- Dyderski, M. K., S. Paż, L. E. Frelich, and A. M. Jagodziński. 2018. How much does climate change threaten European forest tree species distributions? *Global Change Biology* **24**:1150-1163.
- Flousek, J., T. Telenský, J. Hanzelka, and J. Reif. 2015. Population trends of central European montane birds provide evidence for adverse impacts of climate change on high-altitude species. *PLoS One* **10**:e0139465.
- Fraginière, Y., S. Bétrisey, L. Cardinaux, M. Stoffel, and G. Kozłowski. 2015. Fighting their last stand? A global analysis of the distribution and conservation status of gymnosperms. *Journal of Biogeography* **42**:809-820.
- Hofer, S., C. Lang, C. Amory, C. Kittel, A. Delhasse, A. Tedstone, and X. Fettweis. 2020. Greater Greenland Ice Sheet contribution to global sea level rise in CMIP6. *Nature Communications* **11**:1-11.
- IPCC. 2021. Summary for Policymakers. *in* V. Masson-Delmotte, V. P. Zhai, A. Pirani, S. L. Connors, C. Péan, S. Berger, N. Caud, Y. Chen, L. Goldfarb, M. I. Gomis, M. Huang, K. Leitzell, E. Lonnoy, J. B. R. Matthews, T. K. Maycock, T. Waterfield, O. Yelekçi, R. Yu, and B. Zhou, editors. *Climate Change 2021: The Physical Science Basis. Contribution of Working Group I to the Sixth Assessment Report of the Intergovernmental Panel on Climate Change*, In press.
- Jackson, L., R. Kahana, T. Graham, M. Ringer, T. Woollings, J. Mecking, and R. Wood. 2015. Global and European climate impacts of a slowdown of the AMOC in a high resolution GCM. *Climate dynamics* **45**:3299-3316.

- Joppa, L. N., D. L. Roberts, and S. L. Pimm. 2011. How many species of flowering plants are there? *Proceedings of the Royal Society B: Biological Sciences* **278**:554-559.
- Liu, J. and P. H. Raven. 2010. China's environmental challenges and implications for the world. *Critical Reviews in Environmental Science and Technology* **40**:823-851.
- Martorell, C., D. M. Montañana, C. Ureta, and M. C. Mandujano. 2015. Assessing the importance of multiple threats to an endangered globose cactus in Mexico: cattle grazing, looting and climate change. *Biological Conservation* **181**:73-81.
- Mueller, G. M. and J. P. Schmit. 2007. Fungal biodiversity: what do we know? What can we predict? *Biodiversity and conservation* **16**:1-5.
- Myers, N., R. A. Mittermeier, C. G. Mittermeier, G. A. Da Fonseca, and J. Kent. 2000. Biodiversity hotspots for conservation priorities. *Nature* **403**:853-858.
- Naranjo-Ortiz, M. A. and T. Gabaldón. 2019. Fungal evolution: Diversity, taxonomy and phylogeny of the Fungi. *Biological Reviews* **94**:2101-2137.
- Stevens, P. F. 2001. Angiosperm Phylogeny Website. Version 14, July 2017 [and more or less continuously updated since]. <http://www.mobot.org/MOBOT/research/APweb/>.
- Suggitt, A. J., R. J. Wilson, N. J. Isaac, C. M. Beale, A. G. Auffret, T. August, J. J. Bennie, H. Q. Crick, S. Duffield, and R. Fox. 2018. Extinction risk from climate change is reduced by microclimatic buffering. *Nature Climate Change* **8**:713-717.
- Sutton, R. T. 2018. ESD Ideas: a simple proposal to improve the contribution of IPCC WGI to the assessment and communication of climate change risks. *Earth System Dynamics* **9**:1155-1158.
- Swingedouw, D., C. B. Rodehacke, S. M. Olsen, M. Menary, Y. Gao, U. Mikolajewicz, and J. Mignot. 2015. On the reduced sensitivity of the Atlantic overturning to Greenland ice sheet melting in projections: a multi-model assessment. *Climate dynamics* **44**:3261-3279.
- Urbani, F., P. D'Alessandro, and M. Biondi. 2017. Using Maximum Entropy Modeling (MaxEnt) to predict future trends in the distribution of high altitude endemic insects in response to climate change. *Bulletin of Insectology* **70**:189-200.
- Ureta, C., C. Martorell, J. Hortal, and J. Fornoni. 2012. Assessing extinction risks under the combined effects of climate change and human disturbance through the analysis of life-history plasticity. *Perspectives in Plant Ecology, Evolution and Systematics* **14**:393-401.
- Velasco, J. A., F. Estrada, O. Calderón-Bustamante, D. Swingedouw, C. Ureta, C. Gay, and D. Defrance. 2021. Synergistic impacts of global warming and thermohaline circulation collapse on amphibians. *Communications Biology* **4**:1-7.
- WWF. 2020. Living planet report.

Reviewers' comments:

Reviewer #1 (Remarks to the Author):

The revised version of the manuscript is substantially improved.

All comments by the reviewers have been answered appropriately. Results are nicely discussed by the authors and I have nothing to add to this version.

Reviewer #3 (Remarks to the Author):

I have revised again the manuscript and I do not think it is ready for publication yet. I recommend the authors to work further in the text.

Regarding my previous review, I am not satisfied with the answer to my second main objection (n^o2). My objection was not related to how the multicollinearity problem was solved for each modelled species. So the authors' explanation about how the analysis to reduce correlated variables was carried out does not solve my question.

Ecological niche models need a previous selection of ecologically meaningful variables. That implies knowing your species and the habitat they live in. If this selection is not done, potentially extraneous variables can add statistical noise to predictive models or obscure the effects of other more important variables.

I still do not understand how this selection was made (if made). In L 422-427 is commented that 2 variables are retrieved, but I understand that it is for informative purposes after modeling the species, not in order to select the variables for calibration.

So if the same variables were used for all species without a previous selection based in their ecological requirements (as it seems), it should be clearly stated. There is a great disparity in the requirements of such a big number of plants and tetrapods species used that might affect the results obtained. The use of not appropriate variables might offer spurious relationships affecting the species distribution models and that shortcoming should, at least, be mentioned.

In my opinion, the structure and writing of the text could be further improved and needs to be revised carefully.

- For example, its current writing is clearly mixing text across sections:
 - in the Introduction, some lines (L 90-93) are mentioning results, while these lines would correspond more with the Discussion section. Besides, most of the last paragraph of the Introduction seems to be more related to Methods (actually some parts are repeated in Methods). In turn, maybe the Introduction is lacking information on the hypotheses to be tested based on previous studies, etc.
 - There are also some sentences in Methods that sounds like Results ones (e.g. L 381-383: "The coefficient of variation of our models goes from 0.054-0.081 meaning that...", again in L 412-415...)
- the Discussion could still be improved:
 - e.g.: Discussion on Impacts on species richness and composition (L 236) is more referred to climatic suitability than species richness... I see many areas in Figure 6 with gain in SR and these results are not discussed.

MINOR CHANGES:

- L 34: it should be remarked here that RCP 8.5 (the only one used) is the most pessimistic scenario

- In which units is the reduction presented in the abstract (and also across the manuscript) expressed? (0.28–0.48, etc...) Is that a percentage? Please, clarify this across sections.

L 92: where have the variables used been described?

L 318: 500 hundred (= 50000) ? Maybe express numbers differently...

L 318: Why the election of this number of randomly selected records? Any reference of a previous study on it?

L 392: I think there is a contradiction between using always 10,000 pseudo-absences and setting the prevalence to 0.5. Given that not all species have the same number of presences, if you fix 10000 PA, the prevalence will vary.

L 418: binary maps are obtained after the process explained in the next section... I do not think it should be mentioned here.

L 422: "A given modeling algorithm". Please, specify that algorithm and clarify that sentence.

L 441: PSH first time in Methods, please describe the abbreviation

Reviewer #3 (Remarks to the Author):

1. I have revised again the manuscript and I do not think it is ready for publication yet. I recommend the authors to work further in the text.

Regarding my previous review, I am not satisfied with the answer to my second main objection (n°2). My objection was not related to how the multicollinearity problem was solved for each modelled species. So the authors' explanation about how the analysis to reduce correlated variables was carried out does not solve my question.

Ecological niche models need a previous selection of ecologically meaningful variables. That implies knowing your species and the habitat they live in. If this selection is not done, potentially extraneous variables can add statistical noise to predictive models or obscure the effects of other more important variables.

I still do not understand how this selection was made (if made). In L 422-427 is commented that 2 variables are retrieved, but I understand that it is for informative purposes after modeling the species, not in order to select the variables for calibration. So if the same variables were used for all species without a previous selection based in their ecological requirements (as it seems), it should be clearly stated. There is a great disparity in the requirements of such a big number of plants and tetrapods species used that might affect the results obtained. The use of not appropriate variables might offer spurious relationships affecting the species distribution models and that shortcoming should, at least, be mentioned.

Answer: Thank you for your comment and your concern. We are sorry we did not understand your comment the first time. It is true that we did not previously select variables for each of the 21 146 species evaluated based on their ecological needs. As it is being stated in our main text, the evaluated species are endemic species from megadiverse countries (many of them developing countries). For many of these species, the only biological information available is the distribution information coming from IUCN polygons. A detailed ecological knowledge of each individual species is not available to make an a priori selection. However; the aim of the study is to have a better understanding of general tendencies within groups and between them. Ecological niche modeling has been proven to have a good performance and predictive power even when the variables selection is by statistic methods and not ecological knowledge (Iturralde-Pólit et al. 2017, Pillet et al. 2022). Also, we believe that the use of scarce ecological knowledge for pre-scanning the data can also induce a bias in the approach, since it would be a subjective choice. Thus, even if motivated for potentially good reasons, it can bias our modelling approach by reducing objectivity in the ways statistical relationships are established. Thus, given this

fact and the very large number of species analyzed, we stick to our choice, but we now mention this choice in our text for clarity and transparency. In our case, for each evaluated species (as it can be seen in our code), a correlation analysis was carried out to eliminate variables that have a strong correlation and consequently would not add extra information to our modeling by the “corrSelect” function of the package fuzzySim (Barbosa 2015).

Also we retrieved the variables that were the most important for the modeling and reported the two most frequently important variables. Given that the algorithms used, test different variables combinations until reaching a set of variables that are most informative for the species distribution, the variables contribution to the modeling might be different between species. Getting to know which variables were consistently important for the species, helps us having a better understanding of what is affecting the species. Variables with important contribution to the modeling, correspond to variables that have importance in biologically realistic assessments of factors governing population persistence (Searcy and Shaffer 2016).

When modeling a large amount of species, the statistical method to pre-select variables has already been proven to be useful (Iturralde-Pólit et al. 2017, Pillet et al. 2022). Also this method has been shown to be powerful when ecological information of a species is scarce and the only information available is its distribution (Martinez-Meyer 2005).

However, to clarify for the reader, we have added in our method section that we did not previously choose climatic variables for each of our studied species based on their ecological requirements, but on statistical analyses.

2. In my opinion, the structure and writing of the text could be further improved and needs to be revised carefully.

- For example, its current writing is clearly mixing text across sections:
 - in the Introduction, some lines (L 90-93) are mentioning results, while these lines would correspond more with the Discussion section. Besides, most of the last paragraph of the Introduction seems to be more related to Methods (actually some parts are repeated in Methods). In turn, maybe the Introduction is lacking information on the hypotheses to be tested based on previous studies, etc.

Answer: Thank you for your comment. We have eliminated in the introduction the lines that corresponded to our results and methods sections. We also tried to make clearer the hypothesis that were tested.

- There are also some sentences in Methods that sounds like Results ones (e.g. L 381-383: “The coefficient of variation of our models goes from 0.054–0.081 meaning that...”, again in L 412-415...)

Answer: We changed it to our Results and Discussion section, thank you for noticing this to us.

• the Discussion could still be improved:

-e.g.: Discussion on Impacts on species richness and composition (L 236) is more referred to climatic suitability than species richness... I see many areas in Figure 6 with gain in SR and these results are not discussed.

Answer: It is true that we write about species climatic suitability, because it is the basis of “species richness maps” and “difference in species richness maps”. Where a greater number of species find suitability is what we assume as species rich areas; and where more species gain or loss climatic suitability is where we assume gains and losses of climatic suitability.

Consequently, in Figure 6, we show two different results. On the left side we can see the richness pattern and how most species rich areas decrease or increase in different countries. On the right side, we have the “difference in species richness maps” that show shifts in species potential distribution. These maps from the right side help visualizing where greater gains or losses of species are expected. In other words, they give insight of the potential species turnover. Consequently, any time we are referring to species gains or losses, we are referring to the “difference in species richness maps”. We have clarified this information in our main text. We also made a general revision through the entire section.

In China, we find both, increases in species rich areas and areas with important gains in species suitability (which are expected to increase species turnover in those sites). We argue that China is a constant exception related to greater seasonality in the country that might increase species adaptability (Canale and Henry 2010), its more temperate climate in comparison to the other countries evaluated (Duan et al. 2016) and its large area with high altitudes (Liu and Raven 2010).

MINOR CHANGES:

- L 34: it should be remarked here that RCP 8.5 (the only one used) is the most pessimistic scenario

Answer: Done.

- In which units is the reduction presented in the abstract (and also across the manuscript) expressed? (0.28–0.48, etc...) Is that a percentage? Please, clarify this across sections.

Answer: Done.

L 92: where have the variables used been described?

Answer: The 19 bioclimatic variables are very commonly used in ecological niche modeling. We have added the reference in which they can be consulted, gave a general description and added them all.

L 318: 500 hundred (= 50000) ? Maybe express numbers differently...

Answer: Thank you for your comment. We had a mistake. We meant 500 records.

L 318: Why the election of this number of randomly selected records? Any reference of a previous study on it?

Answer: We decided to use 500 points as a maximum, given that the greater the amount of records, the greater can be the problems associated with spatial bias into the modeling (Aiello-Lammens et al. 2015). In the case of records coming from IUCN polygons, more

records do also require more computing time and they not necessarily give more information into the modeling given that their point's distribution is quite homogeneous. We have clarified this in our text.

L 392: I think there is a contradiction between using always 10,000 pseudo-absences and setting the prevalence to 0.5. Given that not all species have the same number of presences, if you fix 10000 PA, the prevalence will vary.

Answer: In the BIOMOD package, the prevalence that equals 0.5 means absences will be weighted equally to the presences (i.e. the weighted sum of presence equals the weighted sum of absences (Thuiller et al. 2021)). In other words, that presences and pseudo-absences will weigh the same in the calibration process, in order to obtain most accurate species distributions (Barbet-Massin et al. 2012). Prevalence does not have the same meaning as it does in medicine. We have clarified this in our text.

L 418: binary maps are obtained after the process explained in the next section... I do not think it should be mentioned here.

Answer: We have erased that part from that section and wrote it in our results.

L 422: "A given modeling algorithm". Please, specify that algorithm and clarify that sentence.

Answer: All algorithms used. We have clarified the information in our text and incorporated the algorithms used.

L 441: PSH first time in Methods, please describe the abbreviation

Answer: Done.

References

- Aiello-Lammens, M. E., R. A. Boria, A. Radosavljevic, B. Vilela, and R. P. Anderson. 2015. spThin: an R package for spatial thinning of species occurrence records for use in ecological niche models. *Ecography* **38**:541-545.
- Barbosa, A. M. 2015. fuzzySim: applying fuzzy logic to binary similarity indices in ecology. *Methods in ecology and evolution* **6**:853-858.
- Canale, C. I. and P.-Y. Henry. 2010. Adaptive phenotypic plasticity and resilience of vertebrates to increasing climatic unpredictability. *Climate research* **43**:135-147.
- Duan, R.-Y., X.-Q. Kong, M.-Y. Huang, S. Varela, and X. Ji. 2016. The potential effects of climate change on amphibian distribution, range fragmentation and turnover in China. *PeerJ* **4**:e2185.
- Iturralde-Pólit, P., O. Dangles, S. F. Burneo, and C. N. Meynard. 2017. The effects of climate change on a mega-diverse country: predicted shifts in mammalian species richness and turnover in continental Ecuador. *Biotropica* **49**:821-831.
- Liu, J. and P. H. Raven. 2010. China's environmental challenges and implications for the world. *Critical Reviews in Environmental Science and Technology* **40**:823-851.
- Martinez-Meyer, E. 2005. Climate change and biodiversity: some considerations in forecasting shifts in species' potential distributions. *Biodiversity Informatics* **2**.
- Pillet, M., B. Goettsch, C. Merow, B. Maitner, X. Feng, P. R. Roehrdanz, and B. J. Enquist. 2022. Elevated extinction risk of cacti under climate change. *Nature plants* **8**:366-372.
- Searcy, C. A. and H. B. Shaffer. 2016. Do ecological niche models accurately identify climatic determinants of species ranges? *The American Naturalist* **187**:423-435.

REVIEWERS' COMMENTS:

Reviewer #3 (Remarks to the Author):

This version has improved. The uncertainty related to the variables has finally been acknowledged in Methods, and the rest of comments have been addressed.